# Sifting through the Noise: Universal First-Order Methods for Stochastic Variational Inequalities

**Kimon Antonakopoulos**
Univ. Grenoble Alpes, CNRS, Inria, Grenoble INP, LIG 38000 Grenoble, France
kimon.antonakopoulos@inria.fr

**Thomas Pethick**          **Ali Kavis**
École Polytechnique Fédérale de Lausanne (EPFL)
thomas.pethick@epfl.ch    ali.kavis@epfl.ch

**Panayotis Mertikopoulos**
Univ. Grenoble Alpes, CNRS, Inria, Grenoble INP, LIG 38000 Grenoble, France
& Criteo AI Lab
panayotis.mertikopoulos@imag.fr

**Volkan Cevher**
École Polytechnique Fédérale de Lausanne (EPFL)
volkan.cevher@epfl.ch

## Abstract

We examine a flexible algorithmic framework for solving monotone variational inequalities in the presence of randomness and uncertainty. The proposed template encompasses a wide range of popular first-order methods, including dual averaging, dual extrapolation and optimistic gradient algorithms – both adaptive and non-adaptive. Our first result is that the algorithm achieves the optimal rates of convergence for cocoercive problems when the profile of the randomness is known to the optimizer: $\mathcal{O}(1/\sqrt{T})$ for absolute noise profiles, and $\mathcal{O}(1/T)$ for relative ones. Subsequently, we drop all prior knowledge requirements (the absolute/relative variance of the randomness affecting the problem, the operator's cocoercivity constant, etc.), and we analyze an adaptive instance of the method that gracefully interpolates between the above rates – i.e., it achieves $\mathcal{O}(1/\sqrt{T})$ and $(1/T)$ in the absolute and relative cases, respectively. To our knowledge, this is the first universality result of its kind in the literature and, somewhat surprisingly, it shows that an extra-gradient proxy step is not required to achieve optimal rates.

## 1   Introduction

This paper focuses on solving variational inequality problems of the form

$$\text{Find } x^* \in \mathbb{R}^d \text{ such that } \langle A(x^*), x - x^* \rangle \geq 0 \text{ for all } x \in \mathbb{R}^d, \tag{VI}$$

where $A \colon \mathbb{R}^d \to \mathbb{R}^d$ is a *monotone cocoercive* operator, i.e.,

$$\langle A(y) - A(x), y - x \rangle \geq \beta \|A(y) - A(x)\|^2 \quad \text{for some } \beta > 0 \text{ and all } x, y \in \mathbb{R}^d. \tag{CC}$$

The study of variational inequalities is a classical topic in optimization that provides a powerful and elegant unifying framework for a broad spectrum of "convex-structured" problems – including

35th Conference on Neural Information Processing Systems (NeurIPS 2021).

convex minimization, saddle-point problems, and games [6, 13]. In particular, such problems have recently attracted considerable attention in the fields of machine learning (ML) and data science because of their potential applications to generative adversarial networks [16], multi-agent and robust reinforcement learning [40], auction theory [47], and many other areas of interest where the minimization of a single empirical loss function does not suffice.

The golden standard for solving (VI) is provided by first-order methods: these methods can be run with computationally cheap updates that only require (noisy) access to $A$, so they are ideal for problems with very high dimensionality and moderate-to-low precision needs (as is typically the case in ML). More precisely, when $A$ is monotone cocoercive as above, the min-max optimal convergence rate for solving (VI) is $\mathcal{O}(1/T)$ after $T$ oracle calls, and it is achieved by the extra-gradient / mirror-prox algorithm [25, 34] with Polyak-Rupert averaging [42]. However, this method requires access to a *perfect* oracle; if the method is run with an imperfect, *stochastic* first-order oracle, its convergence rate drops to $\mathcal{O}(1/\sqrt{T})$ [23], and this rate cannot be improved without additional assumptions [36].

One case where the $\mathcal{O}(1/\sqrt{T})$ convergence rate *can* be improved is when the underlying operator is *strongly monotone* – i.e., the RHS of (CC) is replaced by $\alpha\|y - x\|^2$ for some $\alpha > 0$. In this case, we can obtain a fast $\mathcal{O}(1/T)$ rate with a rapidly decreasing step-size [17]; however, this acceleration requires knowledge of the strong monotonicity modulus, and there is no known way to adapt to it. In particular, if a stochastic method that has been fine-tuned for strongly monotone operators is run on a merely monotone problem, its rate of convergence suffers a catastrophic drop to $\mathcal{O}(1/\log T)$.

These considerations naturally lead to two key research questions:

1. *Are there any conditions for the method's oracle that would close the stochastic-deterministic convergence gap outlined above?*

2. *Is it possible to design a class of methods that are capable of adapting to the quality of the oracle, and that achieve order-optimal rates without prior knowledge of the problem's parameters?*

**Our contributions in the context of related work.** Our goal in this paper is to provide a range of positive answers to the above questions, both in terms of the required oracle conditions, as well as methods that are able to gracefully interpolate between an $\mathcal{O}(1/T)$ and an $\mathcal{O}(1/\sqrt{T})$ rate depending on the setting at hand.

With regard to the first question, our point of departure is the "relative noise" framework of Polyak [41], in which the variance of the oracle is upper bounded by the square norm of the operator at the queried point. This noise model is particularly relevant in coordinate descent methods for unconstrained problems as well as applications to control theory and signal processing where the operator is calculated based on actual, physical measurements that are only accurate up to a percentage of their true value. In recent applications to ML, this noise model has also been studied in the context of overparametrization [39], representation learning [51], and multi-agent learning [28]. Finally, this oracle model has also been studied under the umbrella of multiplicative noise [21] or growth conditions [8, 45, 48, 50], and it is known to improve the convergence rate of stochastic gradient algorithms with non-adaptive step-sizes, even in non-smooth problems [14].

With regard to the second question, we introduce a flexible first-order algorithmic template that includes as special cases the dual averaging [38], dual extrapolation [37] and optimistic gradient methods [43, 44], and which accounts for both adaptive and non-adaptive variants thereof. Our contributions can then be summarized as follows:

1. For oracles with bounded variance, we show that the proposed methods achieve an $\mathcal{O}(1/\sqrt{T})$ rate of convergence if run with a *non-adaptive*, decreasing step-size.

2. In the relative noise model, this rate improves to $\mathcal{O}(1/T)$, and it is achieved with a *constant* step-size that *does need to be tuned* as a function of $T$.

3. Finally, we provide an *adaptive* step-size rule that allows the method to achieve a fast, $\mathcal{O}(1/T)$ rate under relative noise, and an order-optimal $\mathcal{O}(1/\sqrt{T})$ rate in the absolute noise case.

Importantly, our work shows that an extra-gradient mechanism is *not* required to obtain a fast $\mathcal{O}(1/T)$ rate, as this can be achieved by vanilla dual-averaging methods with a *constant* step-size. This is an elegant consequence of the interplay between cocoercivity and the relative noise model; to the best of our knowledge, the only other work considering these models in tandem is the very recent paper [28].

| | $V_t$ | Lipschitz | | Cocoercive + rel. noise | |
|---|---|---|---|---|---|
| | | Ergodic | Last Iterate | Ergodic | Last Iterate |
| Adapt. dual averaging | $0$ | $1/\sqrt{T}$ [11] | Unknown | $1/T$ | Asym. |
| Adapt. dual extrapolation | $AX_t + \text{rel.noise}$ | $1/\sqrt{T}$ [44] | Unknown | $1/T$ | Asym. |
| Adapt. optimistic gradient | $AX_{t-1/2} + \text{rel.noise}$ | $1/\sqrt{T}$ [12] | Unknown | $1/T$ | Asym. |

**Table 1:** The best known convergence rates in stochastic monotone VIs with our contributions highlighted in gray. *Adaptive* refers to our particular adaptive step-size choice in (Adapt). We obtain various schemes with particular choices of $V_t$. For the nomenclature, please refer to Section 3.2.

Our work closes several open threads in [28], which requires a *vanishing* relative noise level to obtain faster convergence in models with relative noise. A summary of our results in the context of related work can be found in Table 1, and we also elaborate on related work in greater detail in the paper's appendix.

## 2   Problem setup and preliminaries

**Examples and motivation.**   Throughout the sequel, we will focus on solving the variational inequality problem (VI). For completeness (and a certain degree of posterity), we briefly mention some examples below, and we defer to [13, 46] for a panoramic survey of the field.

**Example 1** (Convex Minimization)**.**  If $A = \nabla f$ for some convex function $f$, the solutions of (VI) are precisely the minimizers of $f$.

**Example 2** (Min-Max Problems)**.**  If $A = (\nabla_{x_1} L, -\nabla_{x_2} L)$ for some convex-concave function $L(x_1, x_2)$, then the solutions of (VI) coincide with the (global) saddle points of $L$. More precisely, $x^* = (x_1^*, x_2^*)$ is a solution of (VI) if and only if it holds that

$$L(x_1^*, x_2) \leq L(x_1^*, x_2^*) \leq L(x_1, x_2^*) \ \text{ for all } \ x_1 \in \mathcal{X}_1, x_2 \in \mathcal{X}_2. \tag{SP}$$

In this case, (VI) is sometimes referred to as the "vector field formulation" of (SP).

**Example 3** (Monotone games)**.**  Going beyond the min-max setting, a *continuous game in normal form* is defined as follows: First, consider a finite set of players $\mathcal{N} = \{1, \ldots, N\}$, each with their own action space $\mathcal{X}_i = \mathbb{R}^{d_i}$. During play, each player selects an action $x_i$ from $\mathcal{X}_i$ with the aim of minimizing a loss $\ell_i(x_i; x_{-i})$ determined by the ensemble $x := (x_i; x_{-i}) := (x_1, \ldots, x_N)$ of all players' actions. In this context, a Nash equilibrium is any action profile $x^* \in \mathcal{X}$ that is *unilaterally stable*, i.e.,

$$\ell_i(x_i^*; x_{-i}^*) \leq \ell_i(x_i; x_{-i}^*) \quad \text{for all } x_i \in \mathcal{X}_i \text{ and all } i \in \mathcal{N}. \tag{NE}$$

The corresponding operator associated to the game is $A(x) = (\nabla_{x_i} \ell_i(x_i; x_{-i}))_{i \in \mathcal{N}}$. If $A$ is monotone, then the game is itself called *monotone*, and its Nash equilibria coincide with the solutions of (VI), cf. [7, 13, 27, 30–33, 46] and references therein.

**Regularity conditions.**   As we discussed in the introduction, our blanket regularity assumption for (VI) is that the defining operator $A$ is $\beta$-*cocoercive* in the sense of (CC); for a panoramic overview of cocoercive operators we refer the reader to [6].

Some further comments for the cocoercivity condition are in order. First, one may easily observe that if $A$ is $\beta$-cocoercive, it is also $1/\beta$-Lipschitz. The converse does not hold for the general setting of operators; however, when $A$ is the gradient of a smooth convex function, this is indeed the case [5]. Moreover, even though cocoercivity implies that $A$ is monotone, it does not imply that it is *strictly* monotone – a condition which is usually invoked to ensure the existence and uniqueness of solutions to (VI). Therefore, to avoid pathologies, we make the following assumption for our setting:

**Assumption 1.**  The set $\mathcal{X}^* = \{x^* \in \mathbb{R}^d : x^* \text{ is a solution of (VI)}\}$ is non-empty.

Together with cocoercivity, the existence of a solution will be our only blanket assumption in the sequel.

**The gap function.** With the above setup in hand, a widely used performance measure in order to evaluate a candidate solution of (VI) is the so-called *restricted gap function*:

$$\text{Gap}_{\mathcal{C}}(\hat{x}) = \sup_{x \in \mathcal{C}} \langle A(x), \hat{x} - x \rangle, \tag{Gap}$$

where the "test domain" $\mathcal{C}$ is a non-empty compact subset of $\mathbb{R}^d$. The motivation for this choice of merit function is that it characterizes the solutions of the (VI) via its zeros. Formally, we have the following:

**Proposition 1.** *Let $\mathcal{C}$ be a non-empty convex subset of $\mathbb{R}^d$. Then, the following holds*

1. $\text{Gap}_{\mathcal{C}}(\hat{x}) \geq 0$, *whenever $\hat{x} \in \mathcal{C}$*

2. *If $\text{Gap}_{\mathcal{C}}(\hat{x}) = 0$ and $\mathcal{C}$ contains a neighbourhood of $\hat{x}$, then $\hat{x}$ is a solution of* (VI)

Proposition 1 is a generalization of an earlier characterization by Nesterov [37]; see also [2, 38] and references therein. Moreover, it provides a formal justification for the use of $\text{Gap}_{\mathcal{C}}(\hat{x})$ as a merit function for (VI). To streamline our presentation we defer the proof of the above proposition to the appendix.

## 3 The method

### 3.1 Oracle structure and profiles of randomness

From an algorithmic point of view, in order to solve (VI) we will use iterative methods that require access to a stochastic first-order oracle [36]. Formally, this is a black-box feedback mechanism which, when called at $x$, returns a random dual vector $g(x; \omega)$ with $\omega$ drawn from some (complete) probability space $(\Omega, \mathcal{F}, \mathbb{P})$. In practice, the oracle will be called repeatedly at a (possibly random) sequence of points generated by the algorithm at play. Therefore, once the iterate of the method is generated at each round, the oracle draws an i.i.d. sample $\omega \in \Omega$ and returns a dual vector:

$$g(x; \omega) = A(x) + U(x; \omega) \tag{1}$$

with $U(x; \omega)$ denoting the "measurement error".

In this general setting, we make the following statistical assumptions for the oracle:

**Assumption 2** (Absolute noise). The oracle $g(x; \omega)$ enjoys the following properties:

1. *Almost sure boundedness:* There exists some strictly positive numbers $M > 0$ such that:

$$\|g(x; \omega)\|_* \leq M \quad \text{almost surely} \tag{2}$$

2. *Unbiasedness:* $\mathbb{E}\left[g(x; \omega)\right] = A(x)$

3. *Bounded absolute variance:* $\mathbb{E}\left[\|U(x; \omega)\|_*^2\right] \leq \sigma^2$

Such type of conditions for the oracle are standard, especially in the context of adaptive methods cf. [1, 4, 24, 26]. Also, because the variance of the noise is independent of the value of the operator at the queried point, this type of randomness in the oracle will be called *absolute*.

By contrast, following Polyak [41], the *relative* noise model is defined as follows:

**Assumption 3** (Relative noise). The oracle $g(x; \omega)$ enjoys the following properties:

1. *Almost sure boundedness:* There exists some strictly positive numbers $M > 0$ such that: $\|g(x; \omega)\|_* \leq M$ almost surely

2. *Unbiasedness:* $\mathbb{E}\left[g(x; \omega)\right] = A(x)$

3. *Bounded relative variance:* There exists some positive $c > 0$ such that:

$$\mathbb{E}\left[\|U(x; \omega)\|_*^2\right] \leq c\|A(x)\|_*^2 \tag{3}$$

Assumption 2 is standard for obtaining the typical $\mathcal{O}(1/\sqrt{T})$ convergence rate for stochastic optimization scenarios (see for example [23, 35] and references therein). That said, Assumption 3 will prove itself as the crucial statistical condition that will allow us to recover the well known order-optimal bound $\mathcal{O}(1/T)$ for deterministic settings. For concreteness, we provide an example below:

**Example 4** (Random coordinate descent). Consider a smooth convex function $f$ over $\mathbb{R}^d$, as per Example 1. Then the randomized coordinate descent (RCD) algorithm draws one coordinate $i_t$ at random at each stage, and calculates the partial derivative $v_{i,t} = \partial f / \partial x_{i_t}$. Subsequently, the $i$-th derivative is updated as $X_{i,t+1} = X_{i,t} - d\gamma_t v_{i,t}$.

This update rule can be written in abstract recursive form as $x^+ = x - g(x; \omega)$ where $g_i(x; \omega) = d \cdot \partial f / \partial x_i \cdot \omega$ and $\omega$ is drawn uniformly at random from the set of basis vectors $\{e_1, \ldots, e_d\}$ of $\mathbb{R}^d$. Clearly, $\mathbb{E}[g(x; \omega)] = \nabla f(x)$ by construction; moreover, since $\partial f / \partial x_i = 0$ at the minimum points of $f$, we also have $g(x^*; \omega) = 0$ whenever $x^*$ is a minimizer of $f$ – i.e., the variance of the estimator $g(x; \omega)$ vanishes at the minimum points of $f$. It is then straightforward to verify that $\mathbb{E}[\|\hat{v}(x) - \nabla f(x)\|^2] = \mathcal{O}(\|\nabla f(x)\|^2)$, which is precisely the relative noise condition for $A = \nabla f$.

## 3.2 The methods

We now present the generalized extra-gradient (GEG) family of algorithms. More precisely, given two sequences of dual vectors $V_t$ and $V_{t+1/2}$, (GEG) is given by the following recursive formula:

$$
\begin{aligned}
X_{t+1/2} &= X_t - \gamma_t V_t \\
Y_{t+1} &= Y_t - V_{t+1/2} \\
X_{t+1} &= \gamma_{t+1} Y_{t+1}
\end{aligned}
\tag{GEG}
$$

Heuristically, the machinery behind (GEG) suggests to first generate a leading state $X_{t+1/2}$ by taking a step along $V_t$, then aggregate the vector $V_{t+1/2}$ observed at the leading state by incorporating the second dual sequence $V_{t+1/2}$ and finally update the method by applying a dual averaging step [38, 49]. This idea is well-known in the literature of extra-gradient methods [25, 34, 37]. However, up to this point, we have not assumed anything particular for the sequences of $V_t$ and $V_{t+1/2}$, except that they are dual vectors (but not necessarily queries of a stochastic oracle). This generic choice is the building block that will allow us to include various popular algorithmic schemes and provide a unified framework fo their analysis.

To begin with, we provide the following examples that illustrate the fact that Dual Averaging, Dual Extrapolation and Optimistic Dual Averaging can all be written in the form of (GEG) under different choices of $V_t$ and $V_{t+1/2}$.

**Example 5. Stochastic Dual Averaging [38]:** Consider the case $V_t \equiv 0$ and $V_{t+1/2} \equiv g_{t+1/2} = A(X_{t+1/2}) + U_{t+1/2}$. Then, this yields that $X_{t+1/2} = X_t$ and hence $g_{t+1/2} = g_t = V_{t+1/2}$. Therefore, (GEG) reduces to the dual averaging scheme:

$$
\begin{aligned}
Y_{t+1} &= Y_t - g_t \\
X_{t+1} &= \gamma_{t+1} Y_{t+1}
\end{aligned}
\tag{DA}
$$

**Example 6. Stochastic Dual Extrapolation [37]:** Consider the case now where $V_t \equiv g_t = A(X_t) + U_t$ and $V_{t+1/2} \equiv g_{t+1/2} = A(X_{t+1/2}) + U_{t+1/2}$ are noisy oracle queries at $X_t$ and $X_{t+1/2}$ respectively. Then (GEG) readily yields Nesterov's dual extrapolation method [37]:

$$
\begin{aligned}
X_{t+1/2} &= X_t - \gamma_t g_t \\
Y_{t+1} &= Y_t - g_{t+1/2} \\
X_{t+1} &= \gamma_{t+1} Y_{t+1}
\end{aligned}
\tag{DE}
$$

**Example 7. Stochastic Optimistic Dual Averaging [19, 20, 43, 44]:** Consider the case $V_t \equiv g_{t-1/2} = A(X_{t-1/2}) + U_{t-1/2}$ and $V_{t+1/2} \equiv g_{t+1/2} = A(X_{t+1/2}) + U_{t+1/2}$ are the noisy oracle feedback at $X_{t-1/2}$ and $X_{t+1/2}$ respectively. We then get the optimistic dual averaging method:

$$
\begin{aligned}
X_{t+1/2} &= X_t - \gamma_t g_{t-1/2} \\
Y_{t+1} &= Y_t - g_{t+1/2} \\
X_{t+1} &= \gamma_{t+1} Y_{t+1}
\end{aligned}
\tag{OptDA}
$$

The next crucial step is to provide the key ingredient that will allow us to unify the approach for all algorithms belonging to the family (GEG). This is done by a shared "energy" inequality satisfied by all (GEG)-type schemes. Formally, this is described by the following proposition:

**Proposition 2.** *Assume that $X_t, X_{t+1/2}$ are the iterates of* (GEG) *run with a non-negative, non-increasing step-size $\gamma_t$. Then, for all $x \in \mathbb{R}^d$ the following inequality holds:*

$$\sum_{t=1}^{T} \langle V_{t+1/2}, X_{t+1/2} - x \rangle \leq \frac{\|x\|^2}{2\gamma_{T+1}} + \frac{1}{2} \sum_{t=1}^{T} \gamma_t \|V_{t+1/2} - V_t\|_*^2 - \frac{1}{2} \sum_{t=1}^{T} \frac{1}{\gamma_t} \|X_{t+1/2} - X_t\|^2 \quad (4)$$

Proving Proposition 2 requires tiresome computations, so we defer it to the paper's supplement. Instead, we conclude this section by illustrating the various method-specific template inequalities:

1. (Stochastic Dual Averaging): For $V_{t+1/2} = g_{t+1/2}$ and $V_t = 0$, then (4) becomes:

$$\sum_{t=1}^{T} \langle g_t, X_t - x \rangle \leq \frac{\|x\|^2}{2\gamma_{T+1}} + \frac{1}{2} \sum_{t=1}^{T} \gamma_t \|V_t\|_*^2 \quad (5)$$

2. (Stochastic Dual Extrapolation): For $V_t = g_t$ for all $t = 1, 2, \ldots$ then (4) becomes:

$$\sum_{t=1}^{T} \langle g_{t+1/2}, X_{t+1/2} - x \rangle \leq \frac{\|x\|^2}{2\gamma_{T+1}} + \frac{1}{2} \sum_{t=1}^{T} \gamma_t \|g_{t+1/2} - g_t\|_*^2 - \frac{1}{2} \sum_{t=1}^{T} \frac{1}{\gamma_t} \|X_{t+1/2} - X_t\|^2 \quad (6)$$

3. (Stochastic Optimistic Dual Averaging): For $V_t = g_{t-1/2}$ and $V_{t+1/2} = g_{t+1/2}$ then (4) becomes:

$$\sum_{t=1}^{T} \langle g_{t+1/2}, X_{t+1/2} - x \rangle \leq \frac{\|x\|^2}{2\gamma_{T+1}} + \frac{1}{2} \sum_{t=1}^{T} \gamma_t \|g_{t+1/2} - g_{t-1/2}\|_*^2 - \frac{1}{2} \sum_{t=1}^{T} \frac{1}{\gamma_t} \|X_{t+1/2} - X_t\|^2 \quad (7)$$

## 4 Non-adaptive Analysis

In this section, we derive a series of tight convergence rates for (GEG) under both oracle/noise profiles but with a *non-adaptive* step-size sequences. Due to space constraints, we defer the full analysis to the appendix; however, we provide here a proof sketch of our main results via an appropriate "energy inequality" in Proposition 2.

### 4.1 Absolute random noise

In the context of monotone VIs, assumptions induced by the random oracle model are common and well-understood. Indeed, for the general case of bounded variance, i.e., $\mathbb{E}\left[U_{t+1/2}|\mathcal{F}_{t+1/2}\right] \leq \sigma$, extra-gradient/mirror-prox is known to converge at a rate $\mathcal{O}(1/\sqrt{T})$ [22], with a decreasing step-size of order $\mathcal{O}(1/\sqrt{t})$.

For completeness, we analyze (GEG) under a random oracle profile, i.e., for $V_{t+1/2} = g_{t+1/2} \equiv g(X_{t+1/2}; \omega_{t+1/2})$ satisfying Assumption 2 and $V_t$ being an almost surely bounded sequence of dual vectors. To that end, we employ a decreasing step-size choice, which is summarized in the next theorem.

**Theorem 1.** *Let $X_t, X_{t+1/2}$ be generated by* (GEG) *with a decreasing step-size $\gamma_t = \mathcal{O}(1/\sqrt{t})$. Then, for every compact neighborhood $\mathcal{C} \subset \mathbb{R}_d$ of $x^*$, with $\bar{X}_T = \frac{1}{T} \sum_{t=1}^{T} X_{t+1/2}$, it holds that:*

$$\mathbb{E}\left[\mathrm{Gap}_{\mathcal{C}}\left(\bar{X}_T\right)\right] = \mathcal{O}(1/\sqrt{T}).$$

The arguments for the proof of Theorem 1 are standard and we defer them to the appendix due to space constraints. Thanks to this result, we can now derive the respective method specific rates as special instances. More precisely, we have the following proposition:

**Proposition 3.** *Under Assumption 2 the iterates of* (DA)*,* (DE)*,* (OptDA) *enjoy the following rate:*

$$\mathbb{E}\left[\mathrm{Gap}_{\mathcal{C}}\left(\bar{X}_T\right)\right] = \mathcal{O}(1/\sqrt{T}) \quad (8)$$

## 4.2 Relative random noise

We now turn our attention to the relative random oracle framework, i.e. $V_{t+1/2} = g_{t+1/2}$ satisfying Assumption 2 along with:

$$\mathbb{E}\left[\|V_t\|_*^2|\mathcal{F}_t\right] \leq c\|A(X_t)\|_*^2 \ \text{ for all } \ t = 1, 1/2, \ldots \tag{9}$$

In particular, with a carefully chosen constant step-size, under the additional assumption of relative variance, it is possible to achieve an accelerated rate of $\mathcal{O}(1/T)$. One needs to depart from the standard approach to fully exploit the problem setting, i.e., cocoercivity and relative variance. Essentially, it amounts to ensuring that $\sum_{t=1}^{T}\|A_t\|_*^2$ and $\sum_{t=1}^{T}\|A_{t+1/2}\|_*^2$ are summable. We present our result under the respective setting with a proof sketch that highlights its main ingredients.

**Theorem 2.** *Let $X_t, X_{t+1/2}$ be generated by* (GEG) *with a constant step-size that satisfies*

$$\min\left\{(2L)^{-1}, (4L^2\gamma)^{-1}\right\} - 2\gamma c > 0 \text{ with } L = 1/\beta. \tag{10}$$

*Then, for every compact neighbourhood $\mathcal{C} \subset \mathbb{R}_d$ of $x^*$, with $\bar{X}_T = \frac{1}{T}\sum_{t=1}^{T} X_{t+1/2}$, we have:*

$$\mathbb{E}\left[\mathrm{Gap}_{\mathcal{C}}\left(\bar{X}_T\right)\right] = \mathbb{E}\left[\sup_{X \in \mathcal{C}} \left\langle A(X), \bar{X}_T - X\right\rangle\right] = \mathcal{O}(1/T)$$

*Proof.* With a constant step-size, Proposition 2 implies

$$\sum_{t=1}^{T}\left\langle V_{t+1/2}, X_{t+1/2} - X\right\rangle = \frac{\|X\|^2}{2\gamma} + \frac{\gamma}{2}\sum_{t=1}^{T}\|V_{t+1/2} - V_t\|^2 - \frac{1}{2\gamma}\sum_{t=1}^{T}\|X_t - X_{t+1/2}\|^2$$

We show that using smoothness and cocoercivity of the operator, along with the relative noise condition,

$$\left(\min\left\{(2L)^{-1}, (4L^2\gamma)^{-1}\right\} - 2\gamma c\right)\sum_{t=1}^{T}\left(\mathbb{E}\left[\|A(X_t)\|^2\right] + \mathbb{E}\left[\|A(X_{t+1/2})\|^2\right]\right) \leq \frac{\mathbb{E}\left[\|X\|^2\right]}{\gamma}$$

If constant step-size $\gamma$ satisfies Eq. (10), then there exists some strictly positive real number $\beta$, such that $\mathbb{E}\left[\sum_{t=1}^{T}\left(\|A(X_t)\|^2 + \|A(X_{t+1/2})\|^2\right)\right] \leq \mathbb{E}\left[\|X\|^2/\beta\gamma\right] < +\infty$, which concludes that both $\sum_{t=1}^{T}\|A_t\|_*^2$ and $\sum_{t=1}^{T}\|A_{t+1/2}\|_*^2$ are summable. Using the same arguments as in the proof of Theorem 1, we obtain an upper bound for the gap,

$$\mathbb{E}\left[\mathrm{Gap}_{\mathcal{C}}(\bar{X}_{T+1/2})\right] \leq \frac{\frac{D^2}{2\gamma} + 2\gamma c\sum_{t=1}^{T}\mathbb{E}\left[\|A(X_{t+1/2})\|^2 + \|A(X_t)\|^2\right] + \sqrt{\sum_{t=1}^{T}\mathbb{E}\left[\left\|V_{t+1/2}\right\|_*^2\right]}}{T}.$$

By relative variance and summability of operators,

$$\mathbb{E}\left[\mathrm{Gap}_{\mathcal{C}}(\bar{X}_{T+1/2})\right] = \mathcal{O}(1/T)$$

$\square$

Similar to the setting of absolutely random noise, Theorem 2 implies algorithm-specific convergence bounds, which are presented below:

**Proposition 4.** *Under Assumption 3 the iterates of* (DA)*,* (DE)*,* (OptDA) *enjoy the following rate:*

$$\mathbb{E}\left[\mathrm{Gap}_{\mathcal{C}}\left(\bar{X}_T\right)\right] = \mathcal{O}(1/T) \tag{11}$$

An extra appealing feature of the above is that we are able to derive an asymptotic last iterate trajectory result, i.e., the asymptotic convergence of the iterates themselves before any averaging occurs, almost surely. More precisely, we have the following proposition:

**Proposition 5.** *Under Assumption 3 the iterates of* (DA)*,* (DE)*,* (OptDA) *converge to a* (VI) *solution $x^*$.*

The proof Proposition 5 relies on the fact that the distance of the iterates towards any solution of (VI) is decreasing almost surely along with the fact that the summability of $\|A(X_t)\|_*^2$ guarantees that every limit point of the iterate is also a solution of (VI). To streamline our presentation, we defer the detailed proof to the appendix.

# 5  Adaptive Analysis

By the results of Section 4, one may easily observe the interplay between the $\mathcal{O}(1/\sqrt{T})$ to $\mathcal{O}(1/T)$ convergence rates under different noise profiles and step-sizes policies. Therefore a natural question that arises from this context is the following:

*Can we derive a universal step-size policy that is able to optimally adjust the performance of (GEG) without any prior knowledge of the oracle's noise profile?*

In what follows, this desired property is achieved by running (GEG) with the following adaptive step-size:

$$\gamma_t = \frac{1}{\sqrt{1 + \sum_{j=1}^{t-1} \|V_j - V_{j+1/2}\|_*^2}} \tag{Adapt}$$

The step-size (Adapt) is inspired by [44]; however, in our analysis, we provide a generalized point of view *which does not assume* that $V_t$ necessarily is the oracle query at the respective points as in [44]. This allows us to include in the (Adapt) formulation all the adaptive sttep-sizes typically used for the archetypical schemes introduced in Section 3 . More precisely, we have:

1. *Adaptive Stochastic Dual Averaging:* For $V_t \equiv 0$ (Adapt) becomes the standard AdaNorm stepsize, studied in various works [11, 29]:

$$\gamma_t = \frac{1}{\sqrt{1 + \sum_{j=1}^{t-1} \|g_j\|_*^2}} \tag{12}$$

2. *Adaptive Stochastic Dual Extrapolation:* For $V_t = g_{t+1/2}$ (Adapt) becomes

$$\gamma_t = \frac{1}{\sqrt{1 + \sum_{j=1}^{t-1} \|g_j - g_{j+1/2}\|_*^2}} \tag{13}$$

   as used in, e.g., [3, 44, 47].

3. *Adaptive Stochastic Optimistic Dual Averaging:.* For $V_t = g_{t-1/2}$ (Adapt) becomes the step-size used in [19, 20]:

$$\gamma_t = \frac{1}{\sqrt{1 + \sum_{j=1}^{t-1} \|g_{j+1/2} - g_{j-1/2}\|_*^2}} \tag{14}$$

Section 4, heuristically suggests that the success of $\gamma_t$ should hinge on a simultaneous performance as $1/\sqrt{t}$ for the absolute random oracle feedback and as a constant one. whenever the relative random feedback kicks in. This important interpolation feature is what will show in thhe sequel.

## 5.1  Absolute random noise

We will first treat oracles subject to absolute random nooise. In this case, we have:

**Theorem 3.** *Assume that $X_t, X_{t+1/2}$ are the iterates of (GEG) run with the step-size (Adapt). Then, for every compact neighborhood $\mathcal{C} \subset \mathbb{R}^d$ of a solution $x^*$ of (VI), we have:*

$$\mathbb{E}\left[\mathrm{Gap}_{\mathcal{C}}(\overline{X}_T)\right] = \mathcal{O}(1/\sqrt{T}) \tag{15}$$

*with $\overline{X}_T = (1/T) \sum_{t=1}^{T} X_{t+1/2}$*

As we argued above, the result of Theorem 3 is heuristically justified by the fact that the almost sure boundedness conditions for the sequences:

$$\|V_t\|_* \leq M \text{ almost surely for all } t = 1, 1/2, \ldots \tag{16}$$

yields that $\gamma_t = \Omega(1/\sqrt{t})$. In particular, Theorem 3 yields the following specific convergence guarantees:

**Proposition 6.** *Under Assumption 2 the iterates of (DA), (DE), (OptDA) enjoy the following:*

$$\mathbb{E}\left[Gap_{\mathcal{C}}(\overline{X}_T)\right] = \mathcal{O}(1/\sqrt{T}) \tag{17}$$

## 5.2 Relative random noise

Under the relative random noise conditions, we can obtained the following improved rate $\mathcal{O}(1/T)$ instead of the $\mathcal{O}(1/\sqrt{T})$ rate above. Formally, we have the following result:

**Theorem 4.** *Assume that $X_t, X_{t+1/2}$ are the iterates of* (GEG) *run with the step-size* (Adapt). *Then, for every compact neighborhood $\mathcal{C} \subset \mathbb{R}^d$ of a solution $x^*$ of* (VI), *we have:*

$$\mathbb{E}\left[Gap_{\mathcal{C}}(\overline{X}_T)\right] = \mathcal{O}(1/T) \tag{18}$$

*with $\overline{X}_T = 1/T \sum_{t=1}^{T} X_{t+1/2}$*

The crucial ingredient for the proof of Theorem 4 consists of showing that the adaptive step size stabilizes to a positive constant $\gamma_\infty > 0$. In order to obtain this, the first step is to show that the template inequality of Proposition 2 yields

$$\mathbb{E}\left[\frac{1}{\gamma_{T+1}^2}\right] \leq \left(8c \max\left\{L, 2L^2\right\}\left(\frac{\|x^* - x_1\|^2}{2} + 2G^2 + 1\right) + 1\right)\mathbb{E}\left[\frac{1}{\gamma_{T+1}}\right] \tag{19}$$

Moreover, due to the definition of (Adapt) and Jensen's inequality we have:

$$\mathbb{E}\left[\frac{1}{\gamma_{T+1}}\right] = \mathbb{E}\left[\sqrt{1 + \sum_{t=1}^{T}\|V_t - V_{t+1/2}\|^2}\right] \leq \sqrt{\mathbb{E}\left[1 + \sum_{t=1}^{T}\|V_t - V_{t+1/2}\|^2\right]} = \sqrt{\mathbb{E}\left[\frac{1}{\gamma_{T+1}^2}\right]} \tag{20}$$

Therefore, after combining (19) and (20) we get that $\mathbb{E}\left[\frac{1}{\gamma_{T+1}^2}\right] < +\infty$. This directly implies (by the monotone convergence theorem) that:

$$\frac{1}{\gamma_{T+1}^2} = 1 + \sum_{t=1}^{T}\|V_t - V_{t+1/2}\|_*^2 < +\infty \quad \text{almost surely} \tag{21}$$

which in turn yields that $\sum_{t=1}^{T}\|V_t - V_{t+1/2}\|_*^2$ is summable almost surely. Therefore due to the definition of $\gamma_t$ we have almost surely the following:

$$\gamma_{T+1} = \frac{1}{\sqrt{1 + \sum_{t=1}^{T}\|V_t - V_{t+1/2}\|_*^2}} \rightarrow \frac{1}{\sqrt{1 + \sum_{t=1}^{+\infty}\|V_t - V_{t+1/2}\|_*^2}} = \gamma_\infty > 0 \tag{22}$$

Finally, we conclude by providing the following the respective method specific result. Formally, we have:

**Proposition 7.** *Under Assumption 3 the iterates of* (DA), (DE), (OptDA) *enjoy the following:*

*1. The convergence rate in terms of the restricted gap function for the time-average:*

$$\mathbb{E}\left[Gap_{\mathcal{C}}(\overline{X}_T)\right] = \mathcal{O}(1/T) \tag{23}$$

*2. Their last iterate trajectory converges to a* (VI) *solution $x^*$ almost surely.*

The last iterate convergence result of Proposition 7 refers to the asymptotic convergence of the actual sequences of the methods-before any averaging takes place- and it hinges on the fact that the (random) sequences $\|V_t - V_{t+1/2}\|_*^2$ and $\|A(X_t)\|_*^2$ are summable with probability 1. Having established this, we show that $X_t$ satisfies is a (stochastic) quasi-Fejér sequence [9] (with respect to the solution set $\mathcal{X}^*$) along with the fact that every limit point of $X_t$ belongs to $\mathcal{X}^*$. These two building blocks are sufficient in order to derive the almost sure convergence of the iterate's trajectory.

## 6 Numerical experiments

In this section we validate and explore the consequences of the theoretical results. We adopt the experimental setting considered in [15] which is a particular instance of the Kelly auction with $N = 4$. In its generality in a single resource Kelly auction, there are $N$ players sharing a total

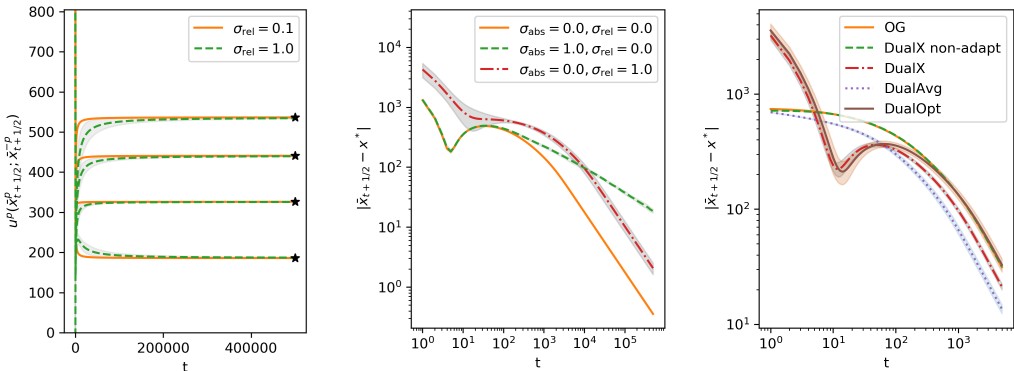

**Figure 1:** (left) Player utility using adaptive DualX for various relative noise levels. Even at relative high levels of noise do we converge to the optimal depicted with ($\star$). (center) Average iterate for deterministic, absolute noise and relative noise using adaptive DualX. We observe the $\mathcal{O}(1/\sqrt{T})$ rate under absolute noise while $\mathcal{O}(1/T)$ is achieved both in the noiseless setting and under relative noise. In addition, as expected, the last iterate only converges under the deterministic and relative noise oracle (see **??**). (right) Average iterate comparing various methods for $\sigma_{\mathrm{rel}} = 0.1$. All methods shares convergence rate with adaptive methods being slightly faster possibly because of difficulty of step-size tuning for non-adaptive methods. Error bars indicate one standard deviation computed using 10 independent executions.

amount of $Q \in \mathbb{R}_{>0}$ resources. At every round, each bidder, $p$, submits a bid $x^p \in \mathbb{R}_{\geq 0}$ and receives proportional resources, $\rho^p = \frac{Qx^p}{Z + \sum_p x^p}$, where $Z$ is the auction entry price. The payoff for player $p$ is then given as $u^p(x^p; x^{-p}) = G^p\rho^p - x^p$, where $G^p$ is the marginal gain in utility for player $p$. One can easily verify that the vectorfield associated with the payoff functions is cocoercive. In addition, the assumption of relative noise can be justified since each player can be seen as performing a measurement when querying the payoff. In such settings, it is common to assume that the error is proportional to the measured quantity and this uncertainty propagates to the gradient information in the form of relative noise. Since players act without communication in this example, it is particularly important that our results extends to single-call extragradient variants (see for instance [44] for elaboration). However, note that our proposed adaptive step-size (Adapt) still relies on global information of all players so our non-adaptive results for known problem constants is also important for this example.

In order to simulate the presence of relative noise we add a term proportional to the norm of the operator. In our notation we can thus capture both relative noise and absolute noise through the error term $U_t$ in the following way,

$$U_t = \epsilon_{\mathrm{rel}}\|A(X_t)\| + \epsilon_{\mathrm{abs}}, \tag{24}$$

where $\epsilon_{\mathrm{rel}} \sim \mathcal{N}(0, \sigma_{\mathrm{rel}}^2)$ and $\epsilon_{\mathrm{abs}} \sim \mathcal{N}(0, \sigma_{\mathrm{abs}}^2)$. To validate the convergence rate we compute the optimal strategy in the deterministic setting (i.e. $\sigma_{\mathrm{rel}} = \sigma_{\mathrm{abs}} = 0$) using Mathematica.

In Fig. 1 we illustrate the behavior of the different instantiations of our algorithmic template under different choices of $\sigma_{\mathrm{rel}}$ and $\sigma_{\mathrm{abs}}$. To denote (DA), (DE) and (OptDA) we use DualAvg, DualX and DualOpt respectively. In addition we include optimistic gradient (OG) from [10] for comparison. For higher dimensional experiments see the appendix, where we additionally apply our adaptive method to the non-convex problem of learning a covariance matrix [10, 18].

## 7 Concluding remarks

In this paper we provide rate interpolation guarantees for different noise profiles; namely that of absolute and relative random noise. That being said our analysis crucially depends on the cocoercivity of the associated operator that defines the respective (VI). It thus remains open whether it is possible to achieve the same $\mathcal{O}(1/T)$ rate for monotone (VI) by only assuming Lipschitz continuity of the said operator and relative noise. Moreover, an additional interesting direction for future research is investigate the impact of relative noise for adaptive accelerated methods and whether it is possible to recover the iconic $\mathcal{O}(1/T^2)$ rate. We postpone these questions to the future.

## Acknowledgments and Disclosure of Funding

This research was partially supported by the COST Action CA16228 "European Network for Game Theory" (GAMENET) and the French National Research Agency (ANR) in the framework of the "Investissements d'avenir" program (ANR-15-IDEX-02), the LabEx PERSYVAL (ANR-11-LABX-0025-01), MIAI@Grenoble Alpes (ANR-19-P3IA-0003), and the grant ALIAS (ANR-19-CE48-0018-01). On the EPFL side, this project has also received funding from the European Research Council (ERC) under the European Union's Horizon 2020 research and innovation programme (grant agreement n° 725594 - time-data), the Hasler Foundation Program: Cyber Human Systems (project number 16066), and the Swiss National Science Foundation (SNSF) under grant number 200021178865/1. This project was sponsored by the Department of the Navy, Office of Naval Research (ONR) under grant number N62909-17-1-2111, and the Army Research Office under Grant Number W911NF-19-1-0404.

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
