\left\| V_{t+1/2} - V_t \right\|_*^2 - \frac{1}{2} \sum_{t=1}^{T} \frac{1}{\gamma_t} \left\| X_{t+1/2} - X_t \right\|^2 \tag{4}$$

Proving Proposition 2 requires tiresome computations, so we defer it to the paper's supplement. Instead, we conclude this section by illustrating the various method-specific template inequalities:

1. (Stochastic Dual Averaging): For $V_{t+1/2} = g_{t+1/2}$ and $V_t = 0$, then (4) becomes:

$$\sum_{t=1}^{T} \langle g_t, X_t - x \rangle \le \frac{\|x\|^2}{2\gamma_{T+1}} + \frac{1}{2} \sum_{t=1}^{T} \gamma_t \left\| V_t \right\|_*^2 \tag{5}$$

2. (Stochastic Dual Extrapolation): For $V_t = g_t$ for all $t = 1, 2, \ldots$ then (4) becomes:

$$\sum_{t=1}^{T} \langle g_{t+1/2}, X_{t+1/2} - x \rangle \le \frac{\|x\|^2}{2\gamma_{T+1}} + \frac{1}{2} \sum_{t=1}^{T} \gamma_t \left\| g_{t+1/2} - g_t \right\|_*^2 - \frac{1}{2} \sum_{t=1}^{T} \frac{1}{\gamma_t} \left\| X_{t+1/2} - X_t \right\|^2 \tag{6}$$

3. (Stochastic Optimistic Dual Averaging): For $V_t = g_{t-1/2}$ and $V_{t+1/2} = g_{t+1/2}$ then (4) becomes:

$$\sum_{t=1}^{T} \langle g_{t+1/2}, X_{t+1/2} - x \rangle \le \frac{\|x\|^2}{2\gamma_{T+1}} + \frac{1}{2} \sum_{t=1}^{T} \gamma_t \left\| g_{t+1/2} - g_{t-1/2} \right\|_*^2 - \frac{1}{2} \sum_{t=1}^{T} \frac{1}{\gamma_t} \left\| X_{t+1/2} - X_t \right\|^2 \tag{7}$$

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

where $\epsilon_{\text{rel}} \sim \mathcal{N}(0, \sigma_{\text{rel}}^2)$ and $\epsilon_{\text{abs}} \sim \mathcal{N}(0, \sigma_{\text{abs}}^2)$. To validate the convergence rate we compute the optimal strategy in the deterministic setting (i.e. $\sigma_{\text{rel}} = \sigma_{\text{abs}} = 0$) using Mathematica.

In Fig. 1 we illustrate the behavior of the different instantiations of our algorithmic template under different choices of $\sigma_{\text{rel}}$ and $\sigma_{\text{abs}}$. To denote (DA), (DE) and (OptDA) we use DualAvg, DualX and DualOpt respectively. In addition we include optimistic gradient (OG) from [12] for comparison. For higher dimensional experiments see the appendix, where we additionally apply our adaptive method to the non-convex problem of learning a covariance matrix [12, 20].

## 7 Concluding remarks

In this paper we provide rate interpolation guarantees for different noise profiles; namely that of absolute and relative random noise. That being said our analysis crucially depends on the cocoercivity of the associated operator that defines the respective (VI). It thus remains open whether it is possible to achieve the same $\mathcal{O}(1/T)$ rate for monotone (VI) by only assuming Lipschitz continuity of the said operator and relative noise. Moreover, an additional interesting direction for future research is investigate the impact of relative noise for adaptive accelerated methods and whether it is possible to recover the iconic $\mathcal{O}(1/T^2)$ rate. We postpone these questions to the future.

## Acknowledgments and Disclosure of Funding

This research was partially supported by the COST Action CA16228 "European Network for Game Theory" (GAMENET) and the French National Research Agency (ANR) in the framework of the "Investissements d'avenir" program (ANR-15-IDEX-02), the LabEx PERSYVAL (ANR-11-LABX-0025-01), MIAI@Grenoble Alpes (ANR-19-P3IA-0003), and the grant ALIAS (ANR-19-CE48-0018-01). On the EPFL side, this project has also received funding from the European Research Council (ERC) under the European Union's Horizon 2020 research and innovation programme (grant agreement n° 725594 - time-data), the Hasler Foundation Program: Cyber Human Systems (project number 16066), and the Swiss National Science Foundation (SNSF) under grant number 200021178865/1. This project was sponsored by the Department of the Navy, Office of Naval Research (ONR) under grant number N62909-17-1-2111, and the Army Research Office under Grant Number W911NF-19-1-0404.

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

# A  Further related work

Due to space limitations for the main paper, we provide in this section a more detailed panorama of the related work.

**Adaptivity**  Adaptive schemes that achieve optimal rates even without knowing the noise constant have been considered before in the min-max optimization setting [4]. However, [4] focuses on the general case where only $\mathcal{O}(1/\sqrt{T})$ is possible. It is also worth pointing out that their step-size relies on the gradient mapping since they consider constrained min-max problems, while ours is based on the operator difference since we consider unconstrained VI problems.

In this sense, [4] is closer to the scheme in [3], where they focus on adapting to non-smooth/smooth problems with unbounded domains in the *deterministic* setting. For the stochastic setting, there exists results for a single-call method using the same adaptive step-size as ours [14]. This work allows us to recover the $\mathcal{O}(T^{-1/2})$ convergence in the general case of Lipschitz operators for the particular instantiation of our algorithmic template.

**Variance reduction**  Another approach to treating the stochasticity is through variance reduction techniques. There is a growing literature on this for variational inequalities [9, 23, 43, 55]. For infinite monotone operators, one approach grows the size of the mini-batches [23], which can be prohibitively expensive in large-scale settings. To this end, [43] exploits a finite sum structure and derive results for strongly monotone operators. Since this approach required knowledge of the problem constant the work, [9] instead relies on a *locally* strongly monotone structure, which is arguably more relevant for non-monotone settings faced in practice. Despite this development variation reduction techniques are known to be brittle to parameter choices possibly explaining their limited use in practice.

**Relative noise**  The assumption of relative noise we rely on dates back to at least Polyak under the name of *relative random noise* [45]. It is a common assumption in the optimization literature but has gone under the guise of various names such as *multiplicative noise* [24]. In particular, for minimization problem it is known as the *growth condition* [8, 49, 52, 54]. This has recently gained interest [16] because of its relationship with the interpolation condition shown to hold for overparameterized models in practice.

**Relative noise in online learning**  In the online learning literature the same noise model that we consider has been studied [31]. This particular noisy feedback model, as it is called in the community, similarly allows them to get finite time last iterate convergence also in the unconstrained setting under cocoercive with unknown constant but for a standard gradient update. However, crucially, they require the relative noise factor to vanish. We get rid of this requirement by employing an extragradient scheme with a different adaptivity, obtaining a $\mathcal{O}(1/T)$-rate for the ergodic average iterate.

# B  Restricted gap function

In this appendix, we discuss the basic properties of the restricted merit function $\mathrm{Gap}_{\mathcal{C}}$ introduced in (Gap). For completeness, we provide the proof of Proposition 1,which itself is an extension of a similar result by [40]:

*Proof of Proposition 1.* Let $x^* \in \mathcal{X}$ be a solution of (VI) so $\langle A(x^*), x - x^* \rangle \geq 0$ for all $x \in \mathcal{X}$. Then, by monotonicity, we get:

$$\langle A(x), x^* - x \rangle \leq \langle A(x) - A(x^*), x^* - x \rangle + \langle A(x^*), x^* - x \rangle$$
$$= -\langle A(x^*) - A(x), x^* - x \rangle - \langle A(x^*), x - x^* \rangle \leq 0, \tag{B.1}$$

so $\mathrm{Gap}_{\mathcal{C}}(x^*) \leq 0$. On the other hand, if $x^* \in \mathcal{C}$, we also get $\mathrm{Gap}(x^*) \geq \langle A(x^*), x^* - x^* \rangle = 0$, so we conclude that $\mathrm{Gap}_{\mathcal{C}}(x^*) = 0$.

For the converse statement, assume that $\mathrm{Gap}_{\mathcal{C}}(\hat{x}) = 0$ for some $\hat{x} \in \mathcal{C}$ and suppose that $\mathcal{C}$ contains a neighborhood of $\hat{x}$ in $\mathcal{X}$. First, we claim that the following inequality holds:

$$\langle A(x), x - \hat{x} \rangle \geq 0 \quad \text{for all } x \in \mathcal{C}. \tag{B.2}$$

Indeed, assume to the contrary that there exists some $x_1 \in \mathcal{C}$ such that

$$\langle A(x_1), x_1 - \hat{x} \rangle < 0. \tag{B.3}$$

This would then give

$$0 = \mathrm{Gap}_{\mathcal{C}}(\hat{x}) \geq \langle A(x_1), \hat{x} - x_1 \rangle > 0, \tag{B.4}$$

which is a contradiction. Now, we further claim that $\hat{x}$ is a solution of (VI),i.e.,:

$$\langle A(\hat{x}), x - \hat{x} \rangle \geq 0 \text{ for all } x \in \mathcal{X}. \tag{B.5}$$

If we suppose that there exists some $z_1 \in \mathcal{X}$ such that $\langle A(\hat{x}), z_1 - \hat{x} \rangle < 0$, then, by the continuity of $A$, there exists a neighborhood $\mathcal{U}'$ of $\hat{x}$ in $\mathcal{X}$ such that

$$\langle A(x), z_1 - x \rangle < 0 \quad \text{for all } x \in \mathcal{U}'. \tag{B.6}$$

Hence, assuming without loss of generality that $\mathcal{U}' \subset \mathcal{U} \subset \mathcal{C}$ (the latter assumption due to the assumption that $\mathcal{C}$ contains a neighborhood of $\hat{x}$), and taking $\lambda > 0$ sufficiently small so that $x = \hat{x} + \lambda(z_1 - \hat{x}) \in \mathcal{U}'$, we get that $\langle A(x), x - \hat{x} \rangle = \lambda \langle A(x), z_1 - \hat{x} \rangle < 0$, in contradiction to (B.2). We conclude that $\hat{x}$ is a solution of (VI), as claimed. □

## C  Template inequalities

In this section we shall provide the proof of the template inequality of Proposition 2. As we already argued in the main. this energy inequality will serve as a template for deriving the method specific convergence rates in the sequel. Formally, we have. the following:

**Proposition 2.** *Assume that $X_t, X_{t+1/2}$ are the iterates of* (GEG) *run with a non-negative, non-increasing step-size $\gamma_t$. Then, for all $x \in \mathbb{R}^d$ the following inequality holds:*

$$\sum_{t=1}^{T} \left\langle V_{t+1/2}, X_{t+1/2} - x \right\rangle \leq \frac{\|x\|^2}{2\gamma_{T+1}} + \frac{1}{2} \sum_{t=1}^{T} \gamma_t \left\| V_{t+1/2} - V_t \right\|_*^2 - \frac{1}{2} \sum_{t=1}^{T} \frac{1}{\gamma_t} \left\| X_{t+1/2} - X_t \right\|^2 \tag{C.1}$$

*Proof.* By the update rule for $X_{t+1}$ in (GEG) we get the following:

$$\begin{aligned}
\left\langle V_{t+1/2}, X_{t+1} - x \right\rangle &= \left\langle \frac{1}{\gamma_t}\gamma_t Y_t - \frac{1}{\gamma_{t+1}}\gamma_{t+1} Y_{t+1}, X_{t+1} - x \right\rangle \\
&= \left\langle \frac{1}{\gamma_t}\gamma_t Y_t - \frac{1}{\gamma_t}\gamma_{t+1}Y_{t+1}, X_{t+1} - x \right\rangle + \left\langle \frac{1}{\gamma_t}\gamma_{t+1}Y_{t+1} - \frac{1}{\gamma_{t+1}}\gamma_{t+1}Y_{t+1}, X_{t+1} - x \right\rangle \\
&= \frac{1}{\gamma_t} \left\langle \gamma_t Y_t - \gamma_{t+1}Y_{t+1}, X_{t+1} - x \right\rangle + \left( \frac{1}{\gamma_{t+1}} - \frac{1}{\gamma_t} \right) \left\langle 0 - \gamma_{t+1}Y_{t+1}, X_{t+1} - x \right\rangle. \\
&= \frac{1}{\gamma_t} \left\langle X_t - X_{t+1}, X_{t+1} - x \right\rangle + \left( \frac{1}{\gamma_{t+1}} - \frac{1}{\gamma_t} \right) \left\langle 0 - X_{t+1}, X_{t+1} - x \right\rangle
\end{aligned}$$

Therefore, by developing the scalar products:

$$\langle X_t - X_{t+1}, X_{t+1} - x \rangle \quad \text{and} \quad \langle 0 - X_{t+1}, X_{t+1} - x \rangle \tag{C.2}$$

we get:

$$\begin{aligned}
\left\langle V_{t+1/2}, X_{t+1} - x \right\rangle &= \frac{1}{\gamma_t} \left[ \frac{1}{2} \|X_{t+1} - x + X_t - X_{t+1}\|^2 - \frac{1}{2} \|X_t - X_{t+1}\|^2 - \frac{1}{2} \|X_{t+1} - x\|^2 \right] \\
&\quad + \left( \frac{1}{\gamma_{t+1}} - \frac{1}{\gamma_t} \right) \left[ \frac{1}{2} \|X_{t+1} - x - X_{t+1}\|^2 - \frac{1}{2} \|X_{t+1}\|^2 - \frac{1}{2} \|X_{t+1} - x\|^2 \right]
\end{aligned} \tag{C.3}$$

which in turn yields:

$$\begin{aligned}
\left\langle V_{t+1/2}, X_{t+1} - x \right\rangle &\leq \frac{1}{2\gamma_t} \|X_t - x\|^2 - \frac{1}{2\gamma_t} \|X_t - X_{t+1}\|^2 - \frac{1}{2\gamma_t} \|X_{t+1} - x\|^2 + \frac{1}{2}\left( \frac{1}{\gamma_{t+1}} - \frac{1}{\gamma_t} \right) \|x\|^2 \\
&\quad - \frac{1}{2\gamma_{t+1}} \|X_{t+1} - x\|^2 + \frac{1}{2\gamma_t} \|X_{t+1} - x\|^2
\end{aligned} \tag{C.4}$$

Therefore, after rearranging,

$$\frac{1}{2\gamma_{t+1}} \|X_{t+1} - x\|^2 \leq \frac{1}{2\gamma_t} \|X_t - x\|^2 + \frac{1}{2}\left(\frac{1}{\gamma_{t+1}} - \frac{1}{\gamma_t}\right) \|x\|^2 - \langle V_{t+1/2}, X_{t+1} - x \rangle - \frac{1}{2\gamma_t} \|X_t - X_{t+1}\|^2$$

$$= \frac{1}{2\gamma_t} \|X_t - x\|^2 + \frac{1}{2}\left(\frac{1}{\gamma_{t+1}} - \frac{1}{\gamma_t}\right) \|x\|^2 - \langle V_{t+1/2}, X_{t+1/2} - x \rangle$$

$$+ \langle V_{t+1/2}, X_{t+1/2} - X_{t+1} \rangle - \frac{1}{2\gamma_t} \|X_t - X_{t+1}\|^2.$$

On the other hand, by invoking the update rule of $X_{t+1/2}$ in (GEG) we have:

$$\gamma_t \langle V_t, X_{t+1/2} - x \rangle = \langle X_t - X_{t+1/2}, X_{t+1/2} - x \rangle$$

$$= \frac{1}{2} \|X_{t+1/2} - x + X_t - X_{t+1/2}\|^2 - \frac{1}{2} \|X_t - X_{t+1/2}\|^2 - \frac{1}{2} \|X_{t+1/2} - x\|^2$$

$$= \frac{1}{2} \|X_t - x\|^2 - \frac{1}{2} \|X_t - X_{t+1/2}\|^2 - \frac{1}{2} \|X_{t+1/2} - x\|^2, \tag{C.5}$$

and after dividing with $\gamma_t$ and rearranging and setting $x = X_{t+1}$

$$\frac{1}{2\gamma_t} \|X_t - X_{t+1/2}\|^2 + \frac{1}{2\gamma_t} \|X_{t+1/2} - X_{t+1}\|^2 + \langle V_t, X_{t+1/2} - X_{t+1} \rangle = \frac{1}{2\gamma_t} \|X_t - X_{t+1}\|^2. \tag{C.6}$$

So, combining the above, we get

$$\frac{1}{2\gamma_{t+1}} \|X_{t+1} - x\|^2 \leq \frac{1}{2\gamma_t} \|X_t - x\|^2 + \frac{1}{2}\left(\frac{1}{\gamma_{t+1}} - \frac{1}{\gamma_t}\right) \|x\|^2 - \langle V_{t+1/2}, X_{t+1/2} - x \rangle$$

$$+ \langle V_{t+1/2}, X_{t+1/2} - X_{t+1} \rangle - \langle V_t, X_{t+1/2} - X_{t+1} \rangle \tag{C.7}$$

$$- \frac{1}{2\gamma_t} \|X_t - X_{t+1/2}\|^2 - \frac{1}{2\gamma_t} \|X_{t+1/2} - X_{t+1}\|^2.$$

Hence, we get:

$$\frac{1}{2\gamma_{t+1}} \|X_{t+1} - x\|^2 \leq \frac{1}{2\gamma_t} \|X_t - x\|^2 - \langle V_{t+1/2}, X_{t+1/2} - x \rangle + \frac{1}{2}\left(\frac{1}{\gamma_{t+1}} - \frac{1}{\gamma_t}\right) \|x\|^2$$

$$+ \underbrace{\langle V_{t+1/2} - V_t, X_{t+1/2} - X_{t+1} \rangle - \frac{1}{2\gamma_t} \|X_{t+1/2} - X_{t+1}\|^2}_{(A)} - \frac{1}{2\gamma_t} \|X_t - X_{t+1/2}\|^2. \tag{C.8}$$

Moreover, by bounding (A) from above we get:

$$\langle V_{t+1/2} - V_t, X_{t+1/2} - X_{t+1} \rangle - \frac{1}{2\gamma_t} \|X_{t+1/2} - X_{t+1}\|^2$$

$$\leq \frac{1}{2}\gamma_t \|V_{t+1/2} - V_t\|_*^2 + \frac{1}{2\gamma_t} \|X_{t+1/2} - X_{t+1}\|^2 - \frac{1}{2\gamma_t} \|X_{t+1/2} - X_{t+1}\|^2 \tag{C.9}$$

$$\leq \frac{1}{2}\gamma_t \|V_{t+1/2} - V_t\|_*^2.$$

So, finally

$$\frac{1}{2\gamma_{t+1}} \|X_{t+1} - x\|^2 \leq \frac{1}{2\gamma_t} \|X_t - x\|^2 - \langle V_{t+1/2}, X_{t+1/2} - x \rangle + \frac{1}{2}\left(\frac{1}{\gamma_{t+1}} - \frac{1}{\gamma_t}\right) \|x\|^2$$

$$+ \frac{1}{2}\gamma_t \|V_t - V_{t+1/2}\|_*^2 - \frac{1}{2\gamma_t} \|X_t - X_{t+1/2}\|^2 \tag{C.10}$$

So, after rearranging and telescoping over $t = 1, \ldots, T$ we get:

$$\sum_{t=1}^{T} \langle V_{t+1/2}, X_{t+1/2} - x \rangle \leq \frac{\|X_1 - x\|^2}{2\gamma_1} + \frac{\|x\|^2}{2\gamma_{T+1}} - \frac{\|x\|^2}{2\gamma_1} + \frac{1}{2}\sum_{t=1}^{T} \gamma_t \|V_t - V_{t+1/2}\|_*^2$$

$$- \frac{1}{2}\sum_{t=1}^{T} \frac{\|X_t - X_{t+1/2}\|^2}{\gamma_t} \tag{C.11}$$

The result follows by setting $X_1 = 0$. $\qquad\square$

We have the following result that will help us to deal with the "noise" martingale difference component.

**Lemma C.1.** *Let $\mathcal{C} \subseteq \mathbb{R}^d$ be a convex set and $h : \mathcal{C} \to \mathbb{R}$ be a 1-strongly-convex with respect to a $\|\cdot\|$ over $\mathcal{C}$. Also, assume that $\forall x \in \mathcal{C}$, $h(x) - \min_{x \in \mathcal{C}} h(x) \leq \frac{D^2}{2}$. Then, for any martingale difference $(Z_t)_{t=1}^T \in \mathbb{R}^d$, and any random vector $x \in \mathcal{C}$, we have:*

$$\mathbb{E}\left[\left\langle \sum_{t=1}^T Z_t, x \right\rangle\right] \leq \frac{\tilde{D}}{2} \sqrt{\sum_{t=1}^T \mathbb{E}\left[\|Z_t\|_*^2\right]} \tag{C.12}$$

The proof of the above lemma could be found in [4], where they present the same result under the label Proposition B.1.

# D    Non-adaptive analysis

*Proof of Theorem 1.* Since we adopt a non-increasing step-size schedule, Proposition 2 applies to this setting. Combining this with almost sure boundedness of stochastic operators,

$$\sum_{t=1}^T \left\langle V_{t+1/2}, X_{t+1/2} - x \right\rangle \leq \frac{\|X\|^2}{2\gamma_{T+1}} + \frac{1}{2} \sum_{t=1}^T \gamma_t \left\|V_{t+1/2} - V_t\right\|_*^2 - \frac{1}{2} \sum_{t=1}^T \frac{1}{\gamma_t} \left\|X_{t+1/2} - X_t\right\|^2$$

$$\leq \frac{\|x\|^2}{2}\sqrt{T+1} + \sum_{t=1}^T \gamma_t \left\|V_{t+1/2}\right\|_*^2 + \gamma_t \left\|V_t\right\|_*^2$$

$$\leq \frac{\|x\|^2}{2}\sqrt{T+1} + 2M^2\sqrt{T}.$$

By monotonicity, and the definition that $V_{t+1/2} = A(X_{t+1/2}) + U_{t+1/2}$,

$$\sum_{t=1}^T \left\langle V_{t+1/2}, X_{t+1/2} - x \right\rangle = \sum_{t=1}^T \left\langle A(X_{t+1/2}), X_{t+1/2} - x \right\rangle + \left\langle U_{t+1/2}, x - X_{t+1/2} \right\rangle$$

$$\geq \sum_{t=1}^T \left\langle A(x), X_{t+1/2} - x \right\rangle + \left\langle U_{t+1/2}, X_{t+1/2} - x \right\rangle$$

$$= T \left\langle A(x), \bar{X}_T - x \right\rangle + \sum_{t=1}^T \left\langle U_{t+1/2}, X_{t+1/2} - x \right\rangle$$

Plugging this lower bound into the first expression,

$$\left\langle A(x), \bar{X}_T - x \right\rangle \leq \frac{\left(\frac{\|x\|^2}{2} + 2M^2\right)\sqrt{T+1} + \sum_{t=1}^T \left\langle U_{t+1/2}, x - X_{t+1/2} \right\rangle}{T}$$

Taking supremum over $x \in \mathcal{C}$ and finally computing expectation with respect to all randomness we obtain

$$\mathbb{E}\left[\mathrm{Gap}_{\mathcal{C}}(\bar{X}_T)\right] \leq \frac{\mathbb{E}\left[\sup_{x \in \mathcal{C}}\left\{\left(\frac{\|x\|^2}{2} + 2M^2\right)\sqrt{T+1} + \underbrace{\sum_{t=1}^T \left\langle U_{t+1/2}, x \right\rangle}_{(A)} - \underbrace{\sum_{t=1}^T \left\langle U_{t+1/2}, X_{t+1/2} \right\rangle}_{(B)}\right\}\right]}{T}.$$

For term (A),

$$\mathbb{E}\left[\sup_{x\in\mathcal{C}}\sum_{t=1}^{T}\left\langle U_{t+1/2},x\right\rangle\right] \leq \mathbb{E}\left[\max_{x\in\mathcal{C}}\left\langle \sum_{t=1}^{T}U_{t+1/2},x\right\rangle\right]$$

$$= \mathbb{E}\left[\left\langle \sum_{t=1}^{T}U_{t+1/2},\tilde{x}\right\rangle\right] \qquad \text{(for some } \tilde{x}\in\mathcal{C} \text{ which attains the maximum)}$$

$$= \frac{\tilde{D}}{2}\sqrt{\sum_{t=1}^{T}\mathbb{E}\left[\left\|U_{t+1/2}\right\|_{*}^{2}\right]} \qquad \text{((by Lemma C.1))}$$

$$= \frac{\tilde{D}}{2}\sqrt{\sum_{t=1}^{T}\mathbb{E}\left[\mathbb{E}\left[\left\|U_{t+1/2}\right\|_{*}^{2}|\mathcal{F}_{t+1/2}\right]\right]}$$

$$= \frac{\tilde{D}}{2}\sigma\sqrt{T} \qquad \text{(Bounded variance)}$$

Also, for term (B),

$$\mathbb{E}\left[\sum_{t=1}^{T}\left\langle U_{t+1/2},X_{t+1/2}\right\rangle\right] = \sum_{t=1}^{T}\mathbb{E}\left[\left\langle U_{t+1/2},X_{t+1/2}\right\rangle\right]$$

$$= \sum_{t=1}^{T}\mathbb{E}\left[\mathbb{E}\left[\left\langle U_{t+1/2},X_{t+1/2}\right\rangle \mid \mathcal{F}_{t+1/2}\right]\right]$$

$$= \sum_{t=1}^{T}\mathbb{E}\left[\left\langle \mathbb{E}\left[U_{t+1/2} \mid \mathcal{F}_{t+1/2}\right],X_{t+1/2}\right\rangle\right]$$

$$= \sum_{t=1}^{T}\mathbb{E}\left[\left\langle 0,X_{t+1/2}\right\rangle\right] \qquad \text{(unbiasedness of } V_{t+1/2})$$

$$= 0.$$

Finally recognizing $\sup_{x\in\mathcal{C}}\|x\| < D$ and combining the expressions for term (A) and (B),

$$\mathbb{E}\left[\text{Gap}_{\mathcal{C}}(\bar{X}_{T})\right] \leq \frac{\mathbb{E}\left[\sup_{x\in\mathcal{C}}\left\{\left(\frac{D^{2}}{2}+2M^{2}\right)\sqrt{T+1}+\frac{\tilde{D}}{2}\sigma\sqrt{T}\right\}\right]}{T},$$

which concludes our derivation

$$\mathbb{E}\left[\text{Gap}_{\mathcal{C}}(\bar{X}_{T})\right] = \mathcal{O}(1/\sqrt{T})$$

$\square$

*Proof of Proposition 3.* Directly obtained by Theorem 1 by setting $V_{t} = 0$ for (DA), $V_{t} = g_{t+1/2}$ for (DE) and $V_{t} = g_{t-1/2}$ for (OptDA).

$\square$

*Proof of Theorem 2.*

$$\sum_{t=1}^{T} \left\langle V_{t+1/2}, X_{t+1/2} - x \right\rangle = \sum_{t=1}^{T} \left\langle V_{t+1/2} - V_t, X_{t+1/2} - X_{t+1} \right\rangle + \left\langle V_t, X_{t+1/2} - X_{t+1} \right\rangle + \left\langle V_{t+1/2}, X_{t+1} - x \right\rangle$$

$$= \sum_{t=1}^{T} \|V_{t+1/2} - V_t\| \|X_{t+1/2} - X_{t+1}\| + \frac{1}{\gamma} \left\langle X_t - X_{t+1/2}, X_{t+1/2} - X_{t+1} \right\rangle + \frac{1}{\gamma} \left\langle \gamma Y_t - X_{t+1}, X_{t+1} - x \right\rangle$$

$$= \sum_{t=1}^{T} \frac{\gamma}{2} \|V_{t+1/2} - V_t\|^2 + \frac{1}{2\gamma} \|X_{t+1/2} - X_{t+1}\|^2$$

$$+ \frac{1}{2\gamma} \left( \|X_t - X\|^2 - \|X_{t+1} - X\|^2 - \|X_t - X_{t+1/2}\|^2 - \|X_{t+1/2} - X_{t+1}\|^2 \right)$$

$$= \frac{\|X_1 - x\|^2}{2\gamma} + \frac{\gamma}{2} \sum_{t=1}^{T} \|V_{t+1/2} - V_t\|^2 - \frac{1}{2\gamma} \sum_{t=1}^{T} \|X_t - X_{t+1/2}\|^2$$

$$= \frac{\|x\|^2}{2\gamma} + \frac{\gamma}{2} \sum_{t=1}^{T} \|V_{t+1/2} - V_t\|^2 - \frac{1}{2\gamma} \sum_{t=1}^{T} \|X_t - X_{t+1/2}\|^2,$$

where we set $X_1 = 0$. At this point the question is how to introduce the relative noise into the analysis such that we show that the stochastic/deterministic operator norms are summable. This would enable us to achieve the anticipated 1/T rate. In other words, we want to show that

$$\mathbb{E}\left[ \sum_{t=1}^{T} \|A(X_{t+1/2})\|^2 \right] < +\infty$$

$$\mathbb{E}\left[ \sum_{t=1}^{T} \|A(X_t)\|^2 \right] < +\infty$$

We take expectation with respect to all randomness and lower bound the left hand side with the norm of the operator using cocoercivity. Setting $x = x^*$, where $x^*$ is a solution of (VI),

$$\mathbb{E}\left[ \sum_{t=1}^{T} \left\langle V_{t+1/2}, X_{t+1/2} - x^* \right\rangle \right] = \mathbb{E}\left[ \sum_{t=1}^{T} \mathbb{E}\left[ \left\langle V_{t+1/2}, X_{t+1/2} - x^* \right\rangle | \mathcal{F}_{t+1/2} \right] \right]$$

$$= \mathbb{E}\left[ \sum_{t=1}^{T} \left\langle \mathbb{E}\left[ V_{t+1/2} | \mathcal{F}_{t+1/2} \right], X_{t+1/2} - x^* \right\rangle \right]$$

$$= \mathbb{E}\left[ \sum_{t=1}^{T} \left\langle A(X_{t+1/2}) - A(x^*), X_{t+1/2} - x^* \right\rangle \right] \quad \text{(Cocoercivity)}$$

$$\geq \frac{1}{L} \mathbb{E}\left[ \sum_{t=1}^{T} \|A(X_{t+1/2})\|^2 \right]$$

Plugging this into the original expression yields

$$\frac{1}{L} \mathbb{E}\left[ \sum_{t=1}^{T} \|A(X_{t+1/2})\|^2 \right] \leq \frac{\|x^*\|^2}{2\gamma} + \frac{\gamma}{2} \sum_{t=1}^{T} \|V_{t+1/2} - V_t\|^2 - \frac{1}{2\gamma} \sum_{t=1}^{T} \|X_t - X_{t+1/2}\|^2$$

With a similar approach,

$$\mathbb{E}\left[\frac{1}{L}\sum_{t=1}^{T}\|A(X_{t+1/2})\|^2 + \frac{1}{2\gamma}\sum_{t=1}^{T}\|X_t - X_{t+1/2}\|^2\right]$$

$$\geq \mathbb{E}\left[\frac{1}{L}\sum_{t=1}^{T}\|A(X_{t+1/2})\|^2 + \frac{1}{2L^2\gamma}\sum_{t=1}^{T}\|A(X_t) - A(X_{t+1/2})\|^2\right]$$

$$\geq \mathbb{E}\left[\min\left\{\frac{1}{2L}, \frac{1}{4L^2\gamma}\right\}\sum_{t=1}^{T} 2\|A(X_{t+1/2})\|^2 + 2\|A(X_t) - A(X_{t+1/2})\|^2\right]$$

$$\geq \mathbb{E}\left[\min\left\{\frac{1}{2L}, \frac{1}{4L^2\gamma}\right\}\sum_{t=1}^{T}\|A(X_t)\|^2\right]$$

Hence,

$$\mathbb{E}\left[\sum_{t=1}^{T}\min\left\{\frac{1}{2L}, \frac{1}{4L^2\gamma}\right\}\|A(X_t)\|^2 + \frac{1}{L}\|A(X_{t+1/2})\|^2\right] \leq \mathbb{E}\left[\frac{\|x^*\|^2}{\gamma} + \gamma\sum_{t=1}^{T}\|V_{t+1/2} - V_t\|^2\right]$$

We now use the relative variance in the expression on the right hand side. Relying on the towering property of expectation,

$$\mathbb{E}\left[\frac{\|x^*\|^2}{\gamma} + \gamma\sum_{t=1}^{T}\|V_{t+1/2} - V_t\|^2\right] \leq \mathbb{E}\left[\frac{\|x^*\|^2}{\gamma} + 2\gamma\sum_{t=1}^{T}\mathbb{E}\left[\|V_{t+1/2}\|^2|\mathcal{F}_{t+1/2}\right] + \mathbb{E}\left[\|V_t\|^2|\mathcal{F}_t\right]\right]$$

$$\leq \mathbb{E}\left[\frac{\|x^*\|^2}{\gamma} + 2\gamma c\sum_{t=1}^{T}\|A(X_{t+1/2})\|^2 + \|A(X_t)\|^2\right]$$

Combining last two expressions together yields

$$\mathbb{E}\left[\sum_{t=1}^{T}\min\left\{\frac{1}{2L}, \frac{1}{4L^2\gamma}\right\}\left(\|A(X_t)\|^2 + \|A(X_{t+1/2})\|^2\right)\right] \leq \mathbb{E}\left[\frac{\|x^*\|^2}{\gamma} + 2\gamma c\sum_{t=1}^{T}\|A(X_{t+1/2})\|^2 + \|A(X_t)\|^2\right]$$

Grouping the same terms on the same side of the inequality,

$$\mathbb{E}\left[\sum_{t=1}^{T}\left(\min\left\{\frac{1}{2L}, \frac{1}{4L^2\gamma}\right\} - 2\gamma c\right)\left(\|A(X_t)\|^2 + \|A(X_{t+1/2})\|^2\right)\right] \leq \mathbb{E}\left[\frac{\|x^*\|^2}{\gamma}\right]$$

As long as $\min\left\{\frac{1}{2L}, \frac{1}{4L^2\gamma}\right\} - 2\gamma c > 0$, we show that sum of operator norms with respect to both sequences are summable.

To obtain the gap, we will decompose $V_{t+1/2}$ into the full operator plus the noise,

$$\sum_{t=1}^{T}\left\langle V_{t+1/2}, X_{t+1/2} - x\right\rangle = \sum_{t=1}^{T}\left\langle A(X_{t+1/2}), X_{t+1/2} - x\right\rangle + \sum_{t=1}^{T}\left\langle U_{t+1/2}, X_{t+1/2} - x\right\rangle$$

$$\geq \sum_{t=1}^{T}\left\langle A(x), X_{t+1/2} - x\right\rangle + \sum_{t=1}^{T}\left\langle U_{t+1/2}, X_{t+1/2} - x\right\rangle$$

(Monotonicity)

$$= T\left\langle A(x), \bar{X}_{t+1/2} - x\right\rangle + \sum_{t=1}^{T}\left\langle U_{t+1/2}, X_{t+1/2} - x\right\rangle$$

Rearranging and incorporating into the original bound,

$$\langle A(x),\ \bar{X}_T - x \rangle$$
$$\leq \frac{1}{T}\left( \frac{\|x\|^2}{2\gamma} + \sum_{t=1}^{T} \frac{\gamma}{2}\|V_{t+1/2} - V_t\|^2 - \frac{1}{2\gamma}\|X_t - X_{t+1/2}\|^2 + \left\langle U_{t+1/2}, x - X_{t+1/2}\right\rangle \right),$$

We take supremum over $x$ to retrieve the gap function and taking expectation,

$$\mathbb{E}\left[\mathrm{Gap}_{\mathcal{C}}(\bar{X}_T)\right]$$
$$\leq \mathbb{E}\left[\sup_{x\in\mathcal{C}}\left\{ \frac{1}{T}\left( \frac{\|x\|^2}{2\gamma} + \sum_{t=1}^{T} \frac{\gamma}{2}\|V_{t+1/2} - V_t\|^2 - \frac{1}{2\gamma}\|X_t - X_{t+1/2}\|^2 + \left\langle U_{t+1/2}, x - X_{t+1/2}\right\rangle \right) \right\} \right]$$
$$\leq \frac{1}{T}\left( \frac{D^2}{2\gamma} + \sum_{t=1}^{T} \mathbb{E}\left[\gamma\|V_{t+1/2}\|^2 + \gamma\|V_t\|^2\right] + \mathbb{E}\left[\sup_{x\in\mathcal{C}}\left\{\left\langle U_{t+1/2}, x\right\rangle\right\}\right] - \mathbb{E}\left[\left\langle U_{t+1/2}, X_{t+1/2}\right\rangle\right] \right)$$
$$\leq \frac{1}{T}\left( \frac{D^2}{2\gamma} + \gamma c\underbrace{\sum_{t=1}^{T} \mathbb{E}\left[\|A(X_{t+1/2})\|^2 + \|A(X_t)\|^2\right]}_{\text{(i)}} + \underbrace{\sum_{t=1}^{T} \mathbb{E}\left[\sup_{x\in\mathcal{C}}\left\{\left\langle U_{t+1/2}, x\right\rangle\right\}\right]}_{\text{(ii)}} - \underbrace{\sum_{t=1}^{T} \mathbb{E}\left[\left\langle U_{t+1/2}, X_{t+1/2}\right\rangle\right]}_{\text{(iii)}} \right),$$

where we define that $\sup_{x\in\mathcal{C}}\|x\| \leq D$ and use relative variance in the last inequality.

For term (i), we have already proven that this particular summation is finite.

For term (ii),

$$\mathbb{E}\left[\sup_{x\in\mathcal{C}}\sum_{t=1}^{T}\left\langle U_{t+1/2}, x\right\rangle\right] \leq \mathbb{E}\left[\max_{x\in\mathcal{C}}\left\langle \sum_{t=1}^{T} U_{t+1/2}, x\right\rangle\right]$$
$$= \mathbb{E}\left[\left\langle \sum_{t=1}^{T} U_{t+1/2}, \tilde{x}\right\rangle\right] \qquad \text{(for some } \tilde{x}\in\mathcal{C} \text{ which attains the maximum)}$$
$$= \frac{\tilde{D}}{2}\sqrt{\sum_{t=1}^{T}\mathbb{E}\left[\left\|U_{t+1/2}\right\|_*^2\right]} \qquad \text{((by Lemma C.1))}$$
$$= \frac{\tilde{D}}{2}\sqrt{\sum_{t=1}^{T}\mathbb{E}\left[\left\|V_{t+1/2} - A(X_{t+1/2})\right\|_*^2\right]} \qquad \text{(unbiasedness of } V_{t+1/2})$$
$$= \frac{\tilde{D}}{2}\sqrt{\sum_{t=1}^{T}\mathbb{E}\left[\left\|V_{t+1/2}\right\|_*^2\right]} \qquad \text{(Towering property)}$$
$$= \frac{\tilde{D}}{2}\sqrt{\sum_{t=1}^{T}\mathbb{E}\left[c\left\|A(X_{t+1/2})\right\|_*^2\right]} < +\infty \qquad \text{(Relative variance)}$$

Finally for term (iii),

$$\mathbb{E}\left[\sum_{t=1}^{T}\langle U_{t+1/2}, X_{t+1/2}\rangle\right] = \sum_{t=1}^{T}\mathbb{E}\left[\langle U_{t+1/2}, X_{t+1/2}\rangle\right]$$

$$= \sum_{t=1}^{T}\mathbb{E}\left[\mathbb{E}\left[\langle U_{t+1/2}, X_{t+1/2}\rangle \mid \mathcal{F}_{t+1/2}\right]\right]$$

$$= \sum_{t=1}^{T}\mathbb{E}\left[\langle\mathbb{E}\left[U_{t+1/2} \mid \mathcal{F}_{t+1/2}\right], X_{t+1/2}\rangle\right]$$

$$= \sum_{t=1}^{T}\mathbb{E}\left[\langle 0, X_{t+1/2}\rangle\right] \qquad\qquad \text{(unbiasedness of } V_{t+1/2})$$

$$= 0.$$

Since we have shown that either the terms are finite or 0, it immediately implies that

$$\mathbb{E}\left[\mathrm{Gap}_{\mathcal{C}}(\bar{X}_T)\right] = \mathcal{O}(1/T)$$

$\square$

*Proof of Proposition 4.* Directly obtained by Theorem 2 by setting $V_t = 0$ for (DA), $V_t = g_{t+1/2}$ for (DE) and $V_t = g_{t-1/2}$ for (OptDA).

$\square$

# E   Adaptive analysis

In this section we shall provide the proof for (GEG) run with adaptive step-sizes for the various noise profiles.Before doing so, we shall present two key building blocks that we will use for our analysis; for both absolute and relative random noise profiles. In particular, we have:

**Lemma E.1** (32, 29). *For all non-negative numbers $\alpha_1, \ldots \alpha_t$, the following inequality holds:*

$$\sqrt{\sum_{t=1}^{T}\alpha_t} \leq \sum_{t=1}^{T}\frac{\alpha_t}{\sqrt{\sum_{i=1}^{t}\alpha_i}} \leq 2\sqrt{\sum_{t=1}^{T}\alpha_t} \qquad\qquad \text{(E.1)}$$

In order to streamline the presentation of our analysis we defer the proof Lemma E.1 to Appendix G along with several variants concerning inequalities of numerical sequences. Having this result at hand, we will start presenting our analysis with the absolute random noise setting.

*Proof of Theorem 3.* Recalling Proposition 2 the following inequality holds:

$$\sum_{t=1}^{T}\langle V_{t+1/2}, X_{t+1/2} - x\rangle \leq \frac{\|x\|^2}{2\gamma_{T+1}} + \frac{1}{2}\sum_{t=1}^{T}\gamma_t\left\|V_{t+1/2} - V_t\right\|_*^2 \qquad\qquad \text{(E.2)}$$

Moreover, by invoking the fact that $V_{t+1/2} = A(X_{t+1/2}) + U_{t+1/2}$ we have that:

$$\sum_{t=1}^{T}\langle A(X_{t+1/2}), X_{t+1/2} - x\rangle \leq \frac{\|x\|^2}{2\gamma_{T+1}} + \frac{1}{2}\sum_{t=1}^{T}\gamma_t\left\|V_{t+1/2} - V_t\right\|_*^2 + \sum_{t=1}^{T}\langle U_{t+1/2}, x - X_{t+1/2}\rangle$$
$$\text{(E.3)}$$

Now, by applying the monotonicity of $A$ we can bound from below the (LHS) as follows:

$$\sum_{t=1}^{T}\langle A(x), X_{t+1/2} - x\rangle \leq \frac{\|x\|^2}{2\gamma_{T+1}} + \frac{1}{2}\sum_{t=1}^{T}\gamma_t\left\|V_{t+1/2} - V_t\right\|_*^2 + \sum_{t=1}^{T}\langle U_{t+1/2}, x - X_{t+1/2}\rangle \quad \text{(E.4)}$$

So, by dividing both sides by $T$, taking suprema on both sides over a compact neighbourhood of a solution $x^*$ and taking expectations:

$$\mathbb{E}\left[\sup_{x \in \mathcal{C}} \langle A(x), \overline{X}_T - x \rangle\right] \leq D^2/2\mathbb{E}\left[\frac{1}{\gamma_{T+1}}\right] + \frac{1}{2}\sum_{t=1}^{T}\mathbb{E}\left[\gamma_t \left\|V_{t+1/2} - V_t\right\|_*^2\right]$$
$$+ \sum_{t=1}^{T}\mathbb{E}\left[\sup_{x \in \mathcal{C}}\langle U_{t+1/2}, x - X_{t+1/2}\rangle\right] \quad \text{(E.5)}$$

which in turn yields:

$$\mathbb{E}\left[\text{Gap}_{\mathcal{C}}(\overline{X}_T)\right] \leq D^2/2\mathbb{E}\left[\frac{1}{\gamma_{T+1}}\right] + \frac{1}{2}\sum_{t=1}^{T}\mathbb{E}\left[\gamma_t \left\|V_{t+1/2} - V_t\right\|_*^2\right] + \sum_{t=1}^{T}\mathbb{E}\left[\sup_{x \in \mathcal{C}}\langle U_{t+1/2}, x - X_{t+1/2}\rangle\right]$$
(E.6)

Therefore, we are left to bound from above the (RHS). We shall do this term by term: For the term $D^2/2\mathbb{E}\left[\frac{1}{\gamma_{T+1}}\right]$ we have:

$$D^2/2\mathbb{E}\left[\frac{1}{\gamma_{T+1}}\right] = D^2/2\mathbb{E}\left[\sqrt{1 + \sum_{t=1}^{T}\|V_t - V_{t+1/2}\|_*^2}\right] \leq D^2/2\sqrt{1 + 4M^2T} \quad \text{(E.7)}$$

with the second inequality being obtained by the fact that $V_t$ is almost surely bounded for all $t = 1, 1/2, \ldots$ . Moreover, for the term $\frac{1}{2}\sum_{t=1}^{T}\mathbb{E}\left[\gamma_t \left\|V_{t+1/2} - V_t\right\|_*^2\right]$ we have:

$$\frac{1}{2}\sum_{t=1}^{T}\mathbb{E}\left[\gamma_t \left\|V_{t+1/2} - V_t\right\|_*^2\right] = \frac{1}{2}\mathbb{E}\left[\sum_{t=1}^{T}(\gamma_t - \gamma_{t+1})\left\|V_{t+1/2} - V_t\right\|_*^2 + \sum_{t=1}^{T}\gamma_{t+1}\left\|V_{t+1/2} - V_t\right\|_*^2\right]$$
$$\leq \frac{1}{2}\left[4M^2\mathbb{E}\left[\cdot\sum_{t=1}^{T}(\gamma_t - \gamma_{t+1})\right] + \mathbb{E}\left[\sum_{t=1}^{T}\gamma_{t+1}\left\|V_{t+1/2} - V_t\right\|_*^2\right]\right]$$
$$\leq \frac{1}{2}\left[4M^2 + \mathbb{E}\left[\sum_{t=1}^{T}\gamma_{t+1}\left\|V_{t+1/2} - V_t\right\|_*^2\right]\right]$$
(E.8)

Now by applying Lemma E.1 we have:

$$\mathbb{E}\left[\sum_{t=1}^{T}\gamma_{t+1}\left\|V_{t+1/2} - V_t\right\|_*^2\right] = \mathbb{E}\left[\sum_{t=1}^{T}\frac{\left\|V_{t+1/2} - V_t\right\|_*^2}{\sqrt{1 + \sum_{j=1}^{t}\left\|V_{j+1/2} - V_j\right\|_*^2}}\right]$$
$$\leq 2\mathbb{E}\left[\sqrt{1 + \sum_{t=1}^{T}\left\|V_{t+1/2} - V_t\right\|_*^2}\right]$$
$$\leq 2\sqrt{1 + 4M^2T}$$

with the last inequality being obtained by the fact that $V_t$ is bounded almost surely for all $t = 1, 1/2, \ldots$ Finally, for the term Bound (B)

$$\mathbb{E}\left[\sup_{x \in \mathcal{C}}\sum_{t=1}^{T}\langle U_{t+1/2}, x - X_{t+1/2}\rangle\right] = \underbrace{\mathbb{E}\left[\sup_{x \in \mathcal{C}}\sum_{t=1}^{T}\langle U_{t+1/2}, x\rangle\right]}_{\text{(B1)}} - \underbrace{\mathbb{E}\left[\sum_{t=1}^{T}\langle U_{t+1/2}, X_{t+1}\rangle\right]}_{\text{(B2)}}$$
(E.9)

For the term (B2) we have:

$$\mathbb{E}\left[\sum_{t=1}^{T}\langle U_{t+1/2}, X_{t+1/2}\rangle\right] = \sum_{t=1}^{T}\mathbb{E}\left[\langle U_{t+1/2}, X_{t+1/2}\rangle\right]$$

$$= \sum_{t=1}^{T}\mathbb{E}\left[\mathbb{E}\left[\langle U_{t+1/2}, X_{t+1/2}\rangle \mid \mathcal{F}_{t+1/2}\right]\right]$$

$$= \sum_{t=1}^{T}\mathbb{E}\left[\langle \mathbb{E}\left[U_{t+1/2} \mid \mathcal{F}_{t+1/2}\right], X_{t+1/2}\rangle\right]$$

$$= \sum_{t=1}^{T}\mathbb{E}\left[\langle 0, X_{t+1/2}\rangle\right] \qquad \text{(unbiasedness of } V_{t+1/2})$$

$$= 0.$$

For the term (B1) we will use Lemma C.1 and we get:

$$\mathbb{E}\left[\sup_{x\in\mathcal{C}}\sum_{t=1}^{T}\langle U_{t+1/2}, x\rangle\right] \leq \mathbb{E}\left[\max_{x\in\mathcal{C}}\left\langle\sum_{t=1}^{T}U_{t+1/2}, x\right\rangle\right]$$

$$= \mathbb{E}\left[\left\langle\sum_{t=1}^{T}U_{t+1/2}, \tilde{x}\right\rangle\right] \qquad \text{(for some } \tilde{x}\in\mathcal{C} \text{ which attains the maximum)}$$

$$\leq \frac{D}{2}\sqrt{\sum_{t=1}^{T}\mathbb{E}\left[\left\|U_{t+1/2}\right\|_*^2\right]} \qquad \text{(by Lemma C.1)}$$

$$\leq \frac{D\sigma}{2}\sqrt{T}$$

Therefore, by combining all the above the result follows. $\square$

Now, we can apply Theorem 3 to directly obtain Proposition 6:

**Proposition 6.** *Under Assumption 2 the iterates of* (DA)*,* (DE)*,* (OptDA) *enjoy the following:*

$$\mathbb{E}\left[Gap_{\mathcal{C}}(\overline{X}_T)\right] = \mathcal{O}(1/\sqrt{T}) \tag{E.10}$$

*Proof.* Directly obtained by Theorem 3 by setting $V_t = 0$ for (DA), $V_t = g_{t+1/2}$ for (DE) and $V_t = g_{t-1/2}$ for (OptDA). $\square$

Now, we turn our attention towards the relative random noise. In particular, in order to show our main results for this context we will the following proposition as a stepping stone. As a prelude, we point out that the following result will also play a crucial role for establishing the last iterate convergence in Appendix F.

**Proposition E.1.** *Assume that $X_t, X_{t+1/2}$ are the iterates of* (GEG) *run with* (Adapt)*. Then, we have:*

$$\mathbb{E}\left[\frac{1}{\gamma_{T+1}^2}\right] = \mathbb{E}\left[1 + \sum_{t=1}^{T}\|V_t - V_{t+1/2}\|_*^2\right] < +\infty \tag{E.11}$$

*and*

$$\mathbb{E}\left[\sum_{t=1}^{T}\|A(X_{t+1/2})\|_*^2\right] < +\infty \tag{E.12}$$

*and*

$$\mathbb{E}\left[\sum_{t=1}^{T}\|A(X_t)\|_*^2\right] < +\infty \tag{E.13}$$

*and*

$$\mathbb{E}\left[\sum_{t=1}^{T}\|X_{t+1/2} - X_t\|^2\right] < +\infty \tag{E.14}$$

*Proof.* Applying Proposition 2 and for $x = x^*$ with $x^*$ being a solution of (VI), we have

$$\sum_{t=1}^{T} \langle V_{t+1/2}, X_{t+1/2} - x^* \rangle \leq \frac{\|x^*\|^2}{2\gamma_{T+1}} + \underbrace{\frac{1}{2} \sum_{t=1}^{T} \gamma_k \left\| V_{t+1/2} - V_t \right\|_*^2}_{(A)} - \frac{1}{2} \sum_{t=1}^{T} \frac{1}{\gamma_t} \left\| X_{t+1/2} - X_t \right\|^2$$

(E.15)

First, we shall bound from above term (A):

$$\frac{1}{2} \sum_{t=1}^{T} \gamma_t \left\| V_{t+1/2} - V_t \right\|_*^2 = \frac{1}{2} \left[ \sum_{t=1}^{T} (\gamma_t - \gamma_{t+1}) \left\| V_{t+1/2} - V_t \right\|_*^2 + \sum_{t=1}^{T} \gamma_{t+1} \left\| V_{t+1/2} - V_t \right\|_*^2 \right]$$

$$\leq \frac{1}{2} \left[ 4G^2 \cdot \sum_{t=1}^{T} (\gamma_t - \gamma_{t+1}) + \sum_{t=1}^{T} \gamma_{t+1} \left\| V_{t+1/2} - V_t \right\|_*^2 \right]$$

$$\leq 2G^2 + \frac{1}{2} \sum_{t=1}^{T} \gamma_{t+1} \left\| V_{t+1/2} - V_t \right\|_*^2$$

$$\leq 2G^2 \sqrt{1 + \sum_{t=1}^{T} \left\| V_{t+1/2} - V_t \right\|_*^2} + \frac{1}{2} \sum_{t=1}^{T} \frac{\left\| V_{t+1/2} - V_t \right\|_*^2}{\sqrt{1 + \sum_{j=1}^{t} \left\| V_{j+1/2} - V_j \right\|_*^2}}$$

$$\leq 2G^2 \sqrt{1 + \sum_{t=1}^{T} \left\| V_{t+1/2} - V_t \right\|_*^2} + 2 \cdot \frac{1}{2} \sqrt{1 + \sum_{t=1}^{T} \left\| V_{t+1/2} - V_t \right\|_*^2}$$

$$= (2G^2 + 1) \sqrt{1 + \sum_{t=1}^{T} \left\| V_{t+1/2} - V_t \right\|_*^2}$$

$$= (2G^2 + 1) \frac{1}{\gamma_{T+1}}$$

(E.16)

So, the above becomes, if we also take expectations on both sides:

$$(B) = \mathbb{E} \left[ \sum_{t=1}^{T} \langle V_{t+1/2}, X_{t+1/2} - x^* \rangle \right] \leq \frac{\|x^*\|^2}{2} \cdot \mathbb{E} \left[ \frac{1}{\gamma_{T+1}} \right] + (2G^2 + 1) \cdot \mathbb{E} \left[ \frac{1}{\gamma_{T+1}} \right]$$

$$- \frac{1}{2} \mathbb{E} \left[ \sum_{t=1}^{T} \frac{1}{\gamma_t} \left\| X_{t+1/2} - X_t \right\|^2 \right]$$

$$= \left[ \frac{\|x^*\|^2}{2} + 2G^2 + 1 \right] \mathbb{E} \left[ \frac{1}{\gamma_{T+1}} \right] - \frac{1}{2} \mathbb{E} \left[ \sum_{t=1}^{T} \frac{1}{\gamma_t} \left\| X_{t+1/2} - X_t \right\|^2 \right]$$

(E.17)

For term (B) we have:

$$\mathbb{E} \left[ \sum_{t=1}^{T} \langle V_{t+1/2}, X_{t+1/2} - x^* \rangle \right] = \sum_{t=1}^{T} \mathbb{E} \left[ \langle V_{t+1/2}, X_{t+1/2} - x^* \rangle \right]$$

$$= \sum_{t=1}^{T} \mathbb{E} \left[ \mathbb{E} \left[ \langle V_{t+1/2}, X_{t+1/2} - x^* \rangle \mid \mathcal{F}_{t+1/2} \right] \right]$$

$$= \sum_{t=1}^{T} \mathbb{E} \left[ \langle \mathbb{E} \left[ V_{t+1/2} \mid \xi_{t+1/2} \right], X_{t+1/2} - x^* \rangle \right]$$

$$= \sum_{t=1}^{T} \mathbb{E} \left[ \langle A \left( X_{t+1/2} \right), X_{t+1/2} - x^* \rangle \right]$$

(E.18)

and since $A$ is $1/L$-cocoercive, we get:

$$\mathbb{E}\left[\sum_{t=1}^{T}\langle V_{t+1/2}, X_{t+1/2} - x^*\rangle\right] \geq \sum_{t=1}^{T}\frac{1}{L}\mathbb{E}\left[\left\|A\left(X_{t+1/2}\right)\right\|_*^2\right] \tag{E.19}$$

Therefore, by combining (E.17) and (E.19) the first inequality that we get is

$$\frac{1}{L}\sum_{t=1}^{T}\mathbb{E}\left[\left\|A\left(X_{t+1/2}\right)\right\|_*^2\right] \leq \left[\frac{\|x^*\|^2}{2} + 2G^2 + 1\right]\mathbb{E}\left[\frac{1}{\gamma_{T+1}}\right] \tag{E.20}$$

Moreover, we have:

$$\frac{1}{L}\sum_{t=1}^{T}\mathbb{E}\left[\left\|A\left(X_{t+1/2}\right)\right\|_*^2\right] \leq \left[\frac{\|x^*\|^2}{2} + 2G^2 + 1\right]\mathbb{E}\left[\frac{1}{\gamma_{T+1}}\right] - \frac{1}{2}\sum_{t=1}^{T}\mathbb{E}\left[\frac{1}{\gamma_t}\left\|X_{t+1/2} - X_t\right\|^2\right] \tag{E.21}$$

and after rearranging and using the fact that $1/\gamma_t \geq 1$ we have

$$(C) = \frac{1}{L}\sum_{t=1}^{T}\mathbb{E}\left[\left\|A\left(X_{t+1/2}\right)\right\|_*^2\right] + \frac{1}{2}\sum_{t=1}^{T}\mathbb{E}\left[\left\|X_{t+1/2} - X_t\right\|^2\right] \leq \left[\frac{\|x^*\|^2}{2} + 2G^2 + 1\right]E\left[\frac{1}{\gamma_{T+1}}\right] \tag{E.22}$$

For the term (C) we will have the following

$$\frac{1}{L}\sum_{t=1}^{T}\mathbb{E}\left[\left\|A\left(X_{t+1/2}\right)\right\|_*^2\right] + \frac{1}{2}\sum_{t=1}^{T}\mathbb{E}\left[\left\|X_{t+1/2} - X_t\right\|^2\right]$$

$$\geq \frac{1}{L}\sum_{t=1}^{T}\mathbb{E}\left[\left\|A\left(X_{t+1/2}\right)\right\|_*^2\right] + \frac{1}{2L^2}\sum_{t=1}^{T}\mathbb{E}\left[\left\|A\left(X_{t+1/2}\right) - A\left(X_t\right)\right\|_*^2\right]$$

$$\geq \min\left\{\frac{1}{L}, \frac{1}{2L^2}\right\}\sum_{t=1}^{T}\mathbb{E}\left[\left\|A\left(X_{t+1/2}\right)\right\|_*^2 + \left\|A\left(X_{t+1/2}\right) - A\left(X_t\right)\right\|_*^2\right]$$

$$= \min\left\{\frac{1}{2L}, \frac{1}{4L^2}\right\}\sum_{t=1}^{T}\mathbb{E}\left[2\left\|A\left(X_{t+1/2}\right)\right\|_*^2 + 2\left\|A\left(X_{t+1/2}\right) - A\left(X_t\right)\right\|_*^2\right]$$

$$\geq \min\left\{\frac{1}{2L}, \frac{1}{4L^2}\right\}\sum_{t=1}^{T}\mathbb{E}\left[\left\|A\left(X_t\right)\right\|_*^2\right] \tag{E.23}$$

So, we get the following inequalities:

$$\frac{1}{L}\sum_{t=1}^{T}\mathbb{E}\left[\left\|A\left(X_{t+1/2}\right)\right\|_*^2\right] \leq \left[\frac{\|x^*\|^2}{2} + 2G^2 + 1\right]\mathbb{E}\left[\frac{1}{\gamma_{T+1}}\right]$$

$$\min\left\{\frac{1}{2L}, \frac{1}{4L^2}\right\}\sum_{t=1}^{T}\mathbb{E}\left[\left\|A\left(X_t\right)\right\|_*^2\right] \leq \left[\frac{\|x^*\|^2}{2} + 2G^2 + 1\right]\mathbb{E}\left[\frac{1}{\gamma_{T+1}}\right] \tag{ineq}$$

and

$$(D) = \frac{1}{L}\sum_{t=1}^{T}\mathbb{E}\left[\left\|A\left(X_{t+1/2}\right)\right\|_*^2\right] + \min\left\{\frac{1}{2L}, \frac{1}{4L^2}\right\}\sum_{t=1}^{T}\mathbb{E}\left[\left\|A\left(X_t\right)\right\|_*^2\right] \leq 2\left[\frac{\|x^*\|^2}{2} + 2G^2 + 1\right]\mathbb{E}\left[\frac{1}{\gamma_{T+1}}\right] \tag{E.24}$$

For term (D) we have:

$$\frac{1}{L}\sum_{t=1}^{T}\mathbb{E}\left[\left\|A\left(X_{t+1/2}\right)\right\|_{*}^{2}\right] + \min\left\{\frac{1}{2L},\frac{1}{4L^{2}}\right\}\sum_{t=1}^{T}\mathbb{E}\left[\left\|A\left(X_{t}\right)\right\|_{*}^{2}\right]$$

$$\geq \min\left\{\frac{1}{2L},\frac{1}{4L^{2}}\right\}\left[\sum_{t=1}^{T}\mathbb{E}\left[\left\|A\left(X_{t+1/2}\right)\right\|_{*}^{2}\right] + \sum_{t=1}^{T}\mathbb{E}\left[\left\|A\left(X_{t}\right)\right\|_{*}^{2}\right]\right]$$

$$\geq \min\left\{\frac{1}{2L},\frac{1}{4L^{2}}\right\}\left[\sum_{t=1}^{T}\frac{1}{c}\mathbb{E}\left[\left\|V_{t+1/2}\right\|_{*}^{2}\right] + \sum_{t=1}^{T}\frac{1}{c}\mathbb{E}\left[\left\|V_{t}\right\|_{*}^{2}\right]\right] \qquad \text{(Assumption 3)}$$

$$\geq \frac{1}{c\max\left\{4L,8L^{2}\right\}}\left[\sum_{t=1}^{T}\mathbb{E}\left[2\left\|V_{t+1/2}\right\|_{*}^{2} + 2\left\|V_{t}\right\|_{*}^{2}\right]\right]$$

$$\geq \frac{1}{c\max\left\{4L,8L^{2}\right\}}\sum_{t=1}^{T}\mathbb{E}\left[\left\|V_{t+1/2} - V_{t}\right\|_{*}^{2}\right]$$

So we get:

$$(E) = \mathbb{E}\left[\sum_{t=1}^{T}\left\|V_{t+1/2} - V_{t}\right\|_{*}^{2}\right] \leq 8c\max\left\{L,2L^{2}\right\}\cdot\left[\frac{\left\|x^{*}\right\|^{2}}{2} + 2G^{2} + 1\right]\mathbb{E}\left[\frac{1}{\gamma_{T+1}}\right] \quad \text{(E.25)}$$

For the term (E) we have:

$$\mathbb{E}\left[\sum_{t=1}^{T}\left\|V_{t+1/2} - V_{t}\right\|_{*}^{2}\right] = \mathbb{E}\left[\sum_{t=1}^{T}\left\|V_{t+1/2} - V_{t}\right\|_{*}^{2} + 1 - 1\right]$$

$$= \mathbb{E}\left[\sum_{t=1}^{T}\left\|V_{t+1/2} - V_{t}\right\|_{*}^{2} + 1\right] - 1 \qquad \text{(E.26)}$$

$$= \mathbb{E}\left[\frac{1}{\gamma_{T+1}^{2}}\right] - 1$$

Therefore

$$\mathbb{E}\left[\frac{1}{\gamma_{T+1}^{2}}\right] \leq 8c\max\left\{L,2L^{2}\right\}\left[\frac{\left\|x^{*}\right\|^{2}}{2} + 2G^{2} + 1\right]\mathbb{E}\left[\frac{1}{\gamma_{T+1}}\right] + 1$$

$$\leq 8c\max\left\{L,2L^{2}\right\}\left[\frac{\left\|x^{*}\right\|^{2}}{2} + 2G^{2} + 1\right]\mathbb{E}\left[\frac{1}{\gamma_{T+1}}\right] + \mathbb{E}\left[\frac{1}{\gamma_{T+1}}\right] \qquad \text{(E.27)}$$

$$= \left[8c\max\left\{L,2L^{2}\right\}\left[\frac{\left\|x^{*}\right\|^{2}}{2} + 2G^{2} + 1\right] + 1\right]\underbrace{\mathbb{E}\left[\frac{1}{\gamma_{T+1}}\right]}_{\text{(F)}}$$

For term (F) we have

$$\mathbb{E}\left[\frac{1}{\gamma_{T+1}}\right] = \mathbb{E}\left[\sqrt{1 + \sum_{t=1}^{T}\left\|V_{t+1/2} - V_{t}\right\|_{*}^{2}}\right]$$

$$\leq \sqrt{\mathbb{E}\left[1 + \sum_{t=1}^{T}\left\|V_{t+1/2} - V_{t}\right\|_{*}^{2}\right]} = \sqrt{\mathbb{E}\left[\frac{1}{\gamma_{T+1}^{2}}\right]} \qquad \text{(E.28)}$$

So, finally we get:

$$\mathbb{E}\left[\frac{1}{\gamma_{T+1}^{2}}\right] \leq \left[8c\max\left\{L,2L^{2}\right\}\left[\frac{\left\|x^{*}\right\|^{2}}{2} + 2G^{2} + 1\right] + 1\right]\sqrt{\mathbb{E}\left[\frac{1}{\gamma_{T+1}^{2}}\right]} \qquad \text{(E.29)}$$

Hence,

$$\mathbb{E}\left[\frac{1}{\gamma_{T+1}^2}\right] \le \left(8c \max\{L, 2L^2\}\left[\frac{\|x^*\|^2}{2} + 2G^2 + 1\right] + 1\right)^2 \tag{E.30}$$

and the first result follows. The second and third claim is derived directly by combining the first claim with (ineq). Finally, the last summability condition by rearranging (E.21), we get:

$$\frac{1}{2}\sum_{t=1}^{T}\mathbb{E}\left[\frac{1}{\gamma_t}\left\|X_{t+1/2} - X_t\right\|^2\right] \le \left[\frac{\|x^*\|^2}{2} + 2G^2 + 1\right]\mathbb{E}\left[\frac{1}{\gamma_{T+1}}\right] \tag{E.31}$$

and the result by our first summability claim. $\qquad\square$

Finally, we shall present the proof of the main result under relative random noise

*Proof of Theorem 4.* Recalling Proposition 2, we have for all $x \in \mathbb{R}^d$

$$\sum_{t=1}^{T}\left\langle V_{t+1/2}, X_{t+1/2} - x\right\rangle \le \frac{\|x\|^2}{2\gamma_{T+1}} + \sum_{t=1}^{T}\gamma_t\left\|V_{t+1/2} - V_t\right\|_*^2 \tag{E.32}$$

By the definition of $V_{t+1/2} = A\left(X_{t+1/2}\right) + U_{t+1/2}$ the above becomes

$$\underbrace{\sum_{t=1}^{T}\left\langle A\left(X_{t+1/2}\right), X_{t+1/2} - x\right\rangle}_{(A)} + \sum_{t=1}^{T}\left\langle U_{t+1/2}, X_{t+1} - x\right\rangle \le \frac{\|x\|^2}{2\gamma_{T+1}} + \sum_{t=1}^{1}\gamma_1\left\|V_{t+1} - V_t\right\|_*^2$$

$$\tag{E.33}$$

Term (A) due to monotonicity of the operator $A$,

$$\sum_{t=1}^{T}\left\langle A\left(X_{t+1/2}\right), X_{t+1/2} - x\right\rangle \ge \sum_{t=1}^{T}\left\langle A(x), X_{t+1/2} - x\right\rangle \text{ for all } x \in \mathbb{R}^d \tag{E.34}$$

Therefore, the above becomes after rearranging,

$$\sum_{t=1}^{T}\left\langle A(x), X_{t+1/2} - x\right\rangle \le \sum_{t=1}^{T}\left\langle U_{t+1/2}, x - X_{t+1/2}\right\rangle + \frac{\|x\|^2}{2\gamma_{T+1}} + \sum_{t=1}^{T}\gamma_t\left\|V_{t+1/2} - V_t\right\|_*^2 \tag{E.35}$$

and by dividing both sides by $T$

$$\left\langle A(x), \bar{X}_T - x\right\rangle \le \frac{1}{T}\left[\sum_{t=1}^{T}\left\langle U_{t+1/2}, x - X_{t+1/2}\right\rangle + \frac{\|x\|^2}{2\gamma_{T+1}} + \sum_{t=1}^{T}\gamma_t\left\|V_{t+1/2} - V_t\right\|_*^2\right] \tag{E.36}$$

and taking suprema on both sides over $\mathcal{C}$ (defining $D^2 = \sup_{x \in \mathcal{C}}\|x - x_1\|^2$),

$$\text{Gap}_{\mathcal{C}}\left(\bar{X}_T\right) \le \frac{1}{T}\left[\sup_{x \in \mathcal{C}}\sum_{t=1}^{T}\left\langle U_{t+1/2}, x - X_{t+1/2}\right\rangle + \frac{D^2}{2\gamma_{T+1}} + \sum_{t=1}^{T}\gamma_t\left\|V_{t+1/2} - V_t\right\|_*^2\right] \tag{E.37}$$

and taking expectations:

$$\mathbb{E}\left[\text{Gap}_{\mathcal{C}}\left(\bar{X}_T\right)\right] \le \frac{1}{T}\Big[\underbrace{\mathbb{E}\left[\sup_{x \in \mathcal{C}}\sum_{t=1}^{T}\left\langle U_{t+1/2}, x - X_{t+1/2}\right\rangle\right]}_{(B)} + \underbrace{\frac{D^2}{2}\mathbb{E}\left[\frac{1}{\gamma_{T+1}}\right]}_{(C)} + \underbrace{\mathbb{E}\left[\sum_{t=1}^{T}\gamma_t\left\|V_{t+1/2} - V_1\right\|_*^2\right]}_{(D)}\Big]$$

$$\tag{E.38}$$

Now, we shall bound the terms (B), (C), (D) individually. Since (B) is the most tricky one we will leave it last.

Bound (C)

$$\frac{D^2}{2}\mathbb{E}\left[\frac{1}{\gamma_{T+1}}\right] = \frac{D^2}{2}\mathbb{E}\left[\sqrt{1 + \sum_{t=1}^{T}\left\|V_{t+1/2} - V_t\right\|_*^2}\right]$$

$$\leq \frac{D^2}{2}\sqrt{\mathbb{E}\left[1 + \sum_{t=1}^{T}\left\|V_{t+1/2} - V_t\right\|_*^2\right]} \qquad (E.39)$$

$$= \frac{D^2}{2}\sqrt{\mathbb{E}\left[\frac{1}{\gamma_{T+1}^2}\right]}$$

$$< +\infty, \text{ from Proposition E.1.}$$

Bound (D)

$$\mathbb{E}\left[\sum_{t=1}^{T}\gamma_t\left\|V_{t+1/2} - V_t\right\|_*^2\right] = \mathbb{E}\left[\sum_{t=1}^{T}(\gamma_t - \gamma_{t+1})\left\|V_{t+1/2} - V_t\right\|_*^2\right] + \mathbb{E}\left[\sum_{t=1}^{T}\gamma_{t+1}\left\|V_{t+1/2} - V_t\right\|_*^2\right]$$

$$\leq 2G^2 + 2\mathbb{E}\left[\sqrt{1 + \sum_{t=1}^{T}\left\|V_{t+1/2} - V_t\right\|_*^2}\right]$$

$$\leq 2G^2 + 2\sqrt{\mathbb{E}\left[1 + \sum_{t=1}^{T}\left\|V_{t+1/2} - V_t\right\|_*^2\right]}$$

$$\leq 2G^2 + 2\sqrt{\mathbb{E}\left[\frac{1}{\gamma_{T+1}^2}\right]}$$

$$< +\infty, \text{ from Proposition E.1}$$

$$(E.40)$$

Bound (B)

$$\mathbb{E}\left[\sup_{x\in\mathcal{C}}\sum_{t=1}^{T}\left\langle U_{t+1/2}, x - X_{t+1/2}\right\rangle\right] = \underbrace{\mathbb{E}\left[\sup_{x\in\mathcal{C}}\sum_{t=1}^{T}\left\langle U_{t+1/2}, x\right\rangle\right]}_{(B1)} - \underbrace{\mathbb{E}\left[\sum_{t=1}^{T}\left\langle U_{t+1/2}, X_{t+1}\right\rangle\right]}_{(B2)}$$

$$(E.41)$$

By working in the same spirit Theorem 3 for the term (B2) we have:

$$\mathbb{E}\left[\sum_{t=1}^{T}\left\langle U_{t+1/2}, X_{t+1/2}\right\rangle\right] = 0. \qquad (E.42)$$

and for term (B1),

$$\mathbb{E}\left[\sup_{x\in\mathcal{C}}\sum_{t=1}^{T}\left\langle U_{t+1/2}, x\right\rangle\right] \leq \frac{D}{2}\sqrt{\sum_{t=1}^{T}\mathbb{E}\left[\left\|U_{t+1/2}\right\|_*^2\right]} \qquad (E.43)$$

Due to the definition of $V_{t+1/2} = A\left(X_{t+1/2}\right) + U_{t+1/2}$ we have $U_{t+1/2} = A\left(X_{t+1/2}\right) - V_{t+1/2}$. So,

$$\mathbb{E}\left[\sup_{x\in e}\sum_{t=1}^{T}\left\langle U_{t+1/2}, x\right\rangle\right] \leq \frac{D}{2}\sqrt{\sum_{t=1}^{T}\mathbb{E}\left[\left\|A\left(X_{t+1}\right) - V_{t+1/2}\right\|_*^2\right]}$$

$$\leq \frac{D}{2}\sqrt{2\sum_{t=1}^{T}\mathbb{E}\left[\left\|A\left(X_{t+1/2}\right)\right\|_*^2\right] + 2\sum_{t=1}^{T}\mathbb{E}\left[\left\|V_{t+1/2}\right\|_*^2\right]} \qquad (E.44)$$

$$< +\infty, \text{ by Proposition E.1}$$

$\square$

Similar to Proposition 6, Theorem 4 allows us to obtain the following result.

**Proposition 7.** *Under Assumption 3 the iterates of* (DA), (DE), (OptDA) *enjoy the following:*

$$\mathbb{E}\left[Gap_{\mathcal{C}}(\overline{X}_T)\right] = \mathcal{O}(1/T) \tag{E.45}$$

*Proof.* Directly obtained by Theorem 4 by setting $V_t = 0$ for (DA), $V_t = g_{t+1/2}$ for (DE) and $V_t = g_{t-1/2}$ for (OptDA). $\qquad\square$

## F Last iterate analysis

We conclude by showing that the iterates $X_{t+1/2}, X_t$ of (GEG) run with the adaptive step-size policy (Adapt) converge towards some (VI) solution $x^*$ almost surely. In doing so, we will need the following proposition:

**Proposition F.1.** *Let there be a non-empty closed set $F$ and let a sequence $(x_t)_t \in \mathbb{R}^d$. Suppose that for all $z \in F$ there exists $(\beta_t)_t$ sequence of random variables satisfying the following almost surely:*

$$\mathbb{E}\left[\|x_{t+1} - z\|^2 \mid \mathcal{F}_t\right] \leq \|x_t - z\|^2 + \beta_t \tag{F.1}$$

*with $\sum_{t=1}^{\infty} \beta_t < +\infty$ almost surely. Then, the following hold:*

1. $\|x_t - z\|^2$ *converges almost surely.*

2. *If the set of almost sure limit points, i.e.*

$$\hat{\mathcal{X}} = \{\hat{x} \in \mathbb{R}^d : \text{there exists a subsequence } x_{t_n} \to \hat{x} \text{ almost surely}\} \tag{F.2}$$

   *is non-empty and $\hat{\mathcal{X}} \subset F$, then $x_t$ converges almost surely to some random variable $\hat{x} \in F$.*

*Proof.* See [11, Proposition 2.3]. $\qquad\square$

Moreover, we will heavily use the following classical convergence theorem; that of the so-called Monotone Convergence Theorem.

**Proposition F.2** (Monotone Convergence Theorem). *Let $(\Omega, \Sigma, \mu)$ be a measure space and $\mathcal{X} \in \Sigma$. Consider a pointwise non-decreasing sequence $(f_t)_t \cdot (\Sigma, \mathcal{B}_{\mathbb{R}_{>0}})$-measurable non-negative functions: $f_t : \mathcal{X} \to [0, +\infty]$. Set the pointwise limit of the $(f_n)$,*

$$\lim_t f_t(x) = f(x) \tag{F.3}$$

*Then, $f$ is $(\Sigma, \mathcal{B}_{\mathbb{R}_{>0}})$-measurable and*

$$\lim_{t \to +\infty} \int_{\mathcal{X}} f_t d\mu = \int_{\mathcal{X}} f d\mu. \tag{F.4}$$

Having all these at hand, we are now in the position to illustrate the last iterate convergence result for the iterates of (DA)/(DE)/(OptDA). For the ease of presentation we shall provide the generic convenience of the general choice for the $V_{t+1/2}$.

**Proposition F.3.** *The iterates of* (DA)/(DE)/(OptDA) *converge towards a* (VI) *solution $x^*$.*

*Proof.* We are left to show that the iterates $X_{t+1/2}$ satisfies the requirements of Proposition F.1. In particular, invoking Proposition 2 we have:

$$\frac{1}{2\gamma_{t+1}}\|X_{t+1} - x^*\|^2 \leq \frac{1}{2\gamma_t}\|X_t - x^*\|^2 - \langle V_{t+1/2}, X_{t+1/2} - x^* \rangle + \frac{D^2}{2}\left(\frac{1}{\gamma_{t+1}} - \frac{1}{\gamma_t}\right) + \gamma_t \|V_t - V_{t+1/2}\|_*^2 \tag{F.5}$$

with $D^2 = \sup_{x^* \in \mathcal{X}} \|x^*\|^2$. Now, by multiplying both sides with $2\gamma_t$ and using the fact that $\gamma_t$ is non-decreasing and $\gamma_t \leq 1$ we get:

$$\|X_{t+1} - x^*\|^2 \leq \|X_t - x^*\| - \gamma_t \langle V_{t+1/2}, X_{t+1/2} - x^* \rangle + \frac{D^2}{2}\left(\frac{1}{\gamma_{t+1}} - \frac{1}{\gamma_t}\right) + \|V_t - V_{t+1/2}\|_*^2 \tag{F.6}$$

Now, by taking conditional expectations we obtain:

$$\mathbb{E}\left[\|X_{t+1} - x^*\|^2 \mid \mathcal{F}_{t+1/2}\right] \leq \|X_t - x^*\|^2 - \gamma_t \mathbb{E}\left[\langle V_{t+1/2}, X_{t+1/2} - x^* \rangle \mid \mathcal{F}_{t+1/2}\right]$$
$$+ \frac{D^2}{2}\mathbb{E}\left[\left(\frac{1}{\gamma_{t+1}} - \frac{1}{\gamma_t}\right) \mid \mathcal{F}_{t+1/2}\right] + \gamma_t \mathbb{E}\left[\|V_t - V_{t+1}\|_*^2 \mid \mathcal{F}_{t+1/2}\right] \quad \text{(F.7)}$$

since $\gamma_t$ is $\mathcal{F}_{t+1/2}$- measurable. Moreover, we have:

$$\gamma_t \mathbb{E}\left[\langle V_{t+1/2}, X_{t+1/2} - x^* \rangle \mid \mathcal{F}_{t+1/2}\right] = \gamma_t \langle \mathbb{E}[V_{t+1/2}|\mathcal{F}_{t+1/2}], X_{t+1/2} - x^* \rangle \leq 0 \quad \text{(F.8)}$$

since $x^*$ is a solution of (VI) and $V_{t+1/2}$ is an unbiased estimator of $A(X_{t+1/2})$. So, we obtain:

$$\mathbb{E}\left[\|X_{t+1} - x^*\|^2 \mid \mathcal{F}_{t+1/2}\right] \leq \|X_t - x^*\|^2 + \frac{D^2}{2}\mathbb{E}\left[\left(\frac{1}{\gamma_{t+1}} - \frac{1}{\gamma_t}\right) \mid \mathcal{F}_{t+1/2}\right] + \mathbb{E}\left[\|V_t - V_{t+1}\|_*^2 \mid \mathcal{F}_{t+1/2}\right]$$
$$\text{(F.9)}$$

The first step is to show that:

$$\beta_t = \frac{D^2}{2}\mathbb{E}\left[\left(\frac{1}{\gamma_{t+1}} - \frac{1}{\gamma_t}\right) \mid \mathcal{F}_{t+1/2}\right] + \mathbb{E}\left[\|V_t - V_{t+1}\|_*^2 \mid \mathcal{F}_{t+1/2}\right] \quad \text{(F.10)}$$

Indeed we have that:

$$\mathbb{E}\left[\sum_{t=1}^T \beta_t\right] = \frac{D^2}{2}\mathbb{E}\left[\sum_{t=1}^T \left(\frac{1}{\gamma_{t+1}} - \frac{1}{\gamma_t}\right)\right] + \mathbb{E}\left[\sum_{t=1}^T \|V_t - V_{t+1}\|_*^2\right]$$
$$\leq \frac{D^2}{2}\mathbb{E}\left[\frac{1}{\gamma_{T+1}}\right] + \mathbb{E}\left[\frac{1}{\gamma_{T+1}^2}\right]$$
$$\leq \left(\frac{D^2}{2} + 1\right)\mathbb{E}\left[\frac{1}{\gamma_{T+1}^2}\right]$$
$$< +\infty$$

due to Proposition E.1. On the other hand, $\sum_{t=1}^T \beta_t$ is a non-decreasing (random) sequence; therefore is converges almost surely to some random value $\sum_{t=1}^{+\infty} \beta_t \in (0, \infty]$. Assume that $\beta_\infty = +\infty$. Then, by applying Proposition F.2 we get:

$$+\infty = \mathbb{E}\left[\sum_{t=1}^{+\infty} \beta_t\right] = \lim_T \mathbb{E}\left[\sum_{t=1}^T \beta_t\right] < +\infty \quad \text{(F.11)}$$

which is a contradiction. Therefore $\sum_{t=1}^{+\infty} \beta_t < +\infty$ almost surely. Therefore, we are left to show that every almost sure limit point of $X_t$ is a (VI) solution. Let $\hat{x} \in \mathbb{R}^d$ be a limit point of $X_t$. Then, there exists a subsequence $X_{t_n}$ which converges almost surely towards $\hat{x}$. Then, by invoking Proposition E.1 (ii), we have that:

$$\mathbb{E}\left[\sum_{t=1}^T \|A(X_t)\|_*^2\right] < +\infty \quad \text{(F.12)}$$

Therefore by the same reasoning as above, Proposition F.2 ensures that:

$$\sum_{t=1}^T \|A(X_t)\|_*^2 < +\infty \quad \text{almost surely} \quad \text{(F.13)}$$

which yields a fortiori that $\|A(X_t)\|_*^2 \to 0$ almost surely. On the other hand, we have that: $\|A(X_{t_n})\|_* \to \|A(\hat{x})\|_*$. Thus, by limit uniqueness we get that that $\|A(\hat{x})\|_* = 0$, so $\hat{x}$ is a (VI) solution, hence the result follows by Proposition F.1. Finally, in order to show that $X_{t+1/2}$ converges also towards a solution, we shall invoke Proposition E.1 (iii) that:

$$\mathbb{E}\left[\sum_{t=1}^T \|X_t - X_{t+1/2}\|^2\right] < +\infty \quad \text{(F.14)}$$

Hence, by the same reasoning we obtain that:

$$\|X_t - X_{t+1/2}\|^2 \to 0 \quad \text{almost surely} \quad \text{(F.15)}$$

and so our proof is completed. □

# G    Lemmas on numerical sequences

In this appendix, we provide the necessary inequality on numerical sequences that we require for the convergence rate analysis of the previous sections.

This lemma that we present is due to [32] and [29].

**Lemma E.1** (32, 29). *For all non-negative numbers $\alpha_1, \ldots \alpha_t$, the following inequality holds:*

$$\sqrt{\sum_{t=1}^{T} \alpha_t} \leq \sum_{t=1}^{T} \frac{\alpha_t}{\sqrt{\sum_{i=1}^{t} \alpha_i}} \leq 2\sqrt{\sum_{t=1}^{T} \alpha_t} \tag{G.1}$$

*Proof.* We begin, by introducing some necessary notation and set $S = \sum_{t=1}^{T} \alpha_t$ and $x = \alpha_T$.

The first part is proved by induction. The induction base case $T = 1$ straightforwardly holds. Now for the induction step, assume that the lemma holds for $T - 1$ and we will show that it holds for $T$ as well. In particular, we have:

$$\sum_{t=1}^{T} \frac{\alpha_t}{\sqrt{\sum_{i=1}^{t} \alpha_i}} = \sum_{t=1}^{T-1} \frac{\alpha_t}{\sqrt{\sum_{i=1}^{t} \alpha_i}} + \frac{\alpha_T}{\sqrt{\sum_{t=1}^{T} \alpha_t}}$$

$$\geq \sqrt{\sum_{t=1}^{T-1} \alpha_t} + \frac{\alpha_T}{\sqrt{\sum_{t=1}^{T} \alpha_t}} = \sqrt{S - x} + \frac{x}{\sqrt{S}} \tag{G.2}$$

where the first inequality is obtained due to the induction hypothesis. Hence, in order to prove the lemma it is sufficient to show that:

$$\sqrt{S - x} + \frac{x}{\sqrt{S}} \geq \sqrt{S} \tag{G.3}$$

By multiplying both sides by $\sqrt{S}$, we get the following equivalent expression:

$$\sqrt{S^2 - xS} \geq S - x \tag{G.4}$$

whereas after rearranging we obtain the equivalent inequality:

$$x \leq S \tag{G.5}$$

which holds, since $x = \alpha_T \leq \sum_{t=1}^{T} \alpha_t = S$ and hence the LHS inequality is obtained.
Now, the proof of the RHS inequality:

$$\sum_{t=1}^{T} \frac{\alpha_t}{\sqrt{\sum_{i=1}^{t} \alpha_i}} \leq 2\sqrt{\sum_{t=1}^{T} \alpha_t} \tag{G.6}$$

will again be done again via induction. The induction base $T = 1$ holds immediately. Assume that the lemma holds for $T - 1$. We will show that it also holds for $T$. By the induction hypothesis, we get:

$$\sum_{t=1}^{T} \frac{\alpha_t}{\sqrt{\sum_{i=1}^{t} \alpha_i}} \leq 2\sqrt{\sum_{t=1}^{T-1} \alpha_t} + \frac{\alpha_T}{\sqrt{\sum_{t=1}^{T} \alpha_t}} = 2\sqrt{S - x} + \frac{x}{\sqrt{S}} \tag{G.7}$$

where $x = \alpha_T$ and $S = \sum_{t=1}^{T} \alpha_t$ (we highlight once more that $x \leq S$). Taking the derivative of the function $H(x) = 2\sqrt{S - x} + \frac{x}{\sqrt{S}}$ (with respect to $x$), we get that:

$$H'(x) = \frac{1}{\sqrt{S}} - \frac{1}{\sqrt{S - x}} \tag{G.8}$$

becomes negative for all $x \geq 0$. Thus, $H(x) \leq H(0) = 2\sqrt{S}$ and therefore the result follows.    □

# H Numerics

All experiments were performed on a MacBook Pro with a 2.7 GHz Quad-Core Intel Core i7 processor and 16 GB of RAM.

## H.1 Kelly auction

We consider a Kelly auction with number of player $N = 4$, cost of bidding $Z = 100$, resources $Q = 1000$ and marginal utility gains $G = (1.8, 2.0, 2.2, 2.4)$ (see Section 6 for exact definitions). The hyperparameters (the step-size for non-adaptive methods and the diameter for adaptive methods) are not fine-tuned but chosen heuristically based on the sweep in Fig. H.5. When error bars are present they represent one standard deviation based on 10 independent executions. For more information on naming and notation see Section 6.

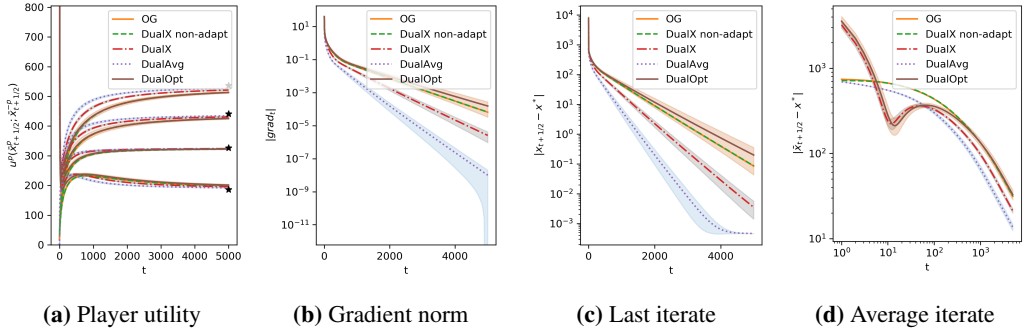

**(a)** Player utility    **(b)** Gradient norm    **(c)** Last iterate    **(d)** Average iterate

**Figure H.1:** Comparing methods with $\sigma_{\mathrm{rel}} = 0.1$.

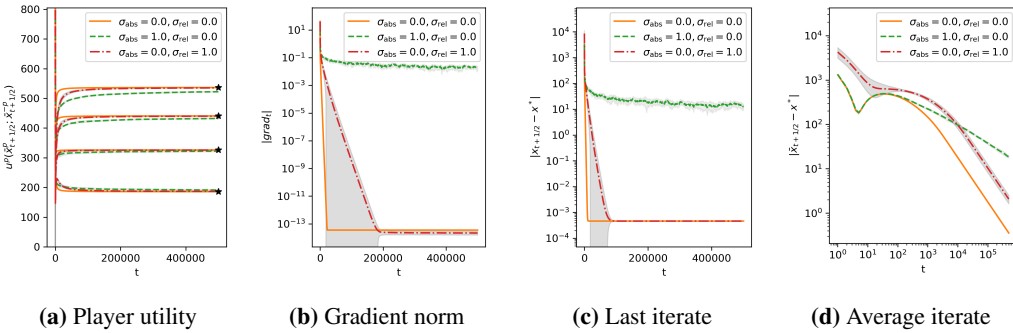

**(a)** Player utility    **(b)** Gradient norm    **(c)** Last iterate    **(d)** Average iterate

**Figure H.2:** Relative noise compared with absolute noise using adaptive DualX. We observe the deterioration of the rate for the average iterate for absolute noise in contrast with relative noise.

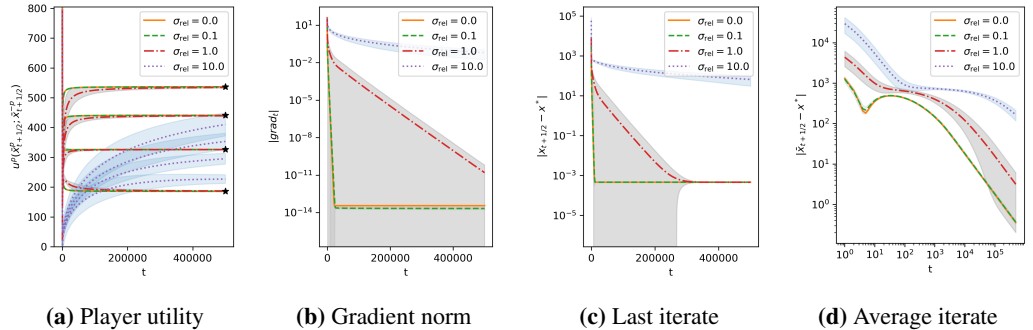

**(a)** Player utility  **(b)** Gradient norm  **(c)** Last iterate  **(d)** Average iterate

**Figure H.3:** Different levels of relative noise using adaptive DualX. The $\mathcal{O}(1/T)$ rate for the average iterate (last plot) is kept even when the noise level is increased to $\sigma_{\text{rel}} = 1.0$.

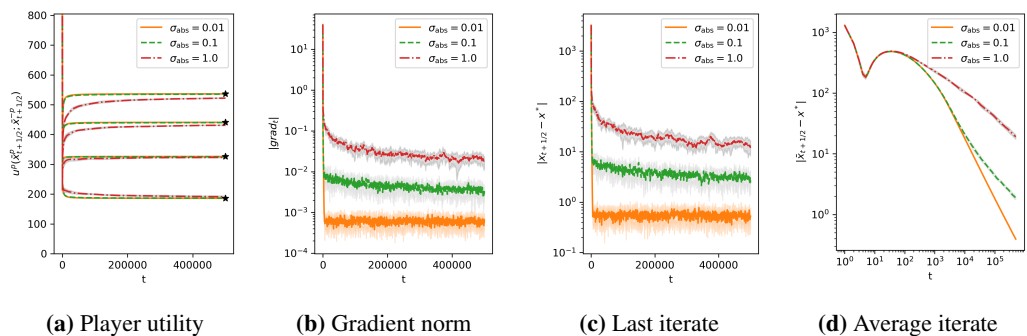

**(a)** Player utility  **(b)** Gradient norm  **(c)** Last iterate  **(d)** Average iterate

**Figure H.4:** Different levels of absolute noise using adaptive DualX. In contrast with relative noise increasing the absolute noise clearly worsens the slope for the average iterate (last plot) indicating a rate of $\mathcal{O}(1/\sqrt{T})$.

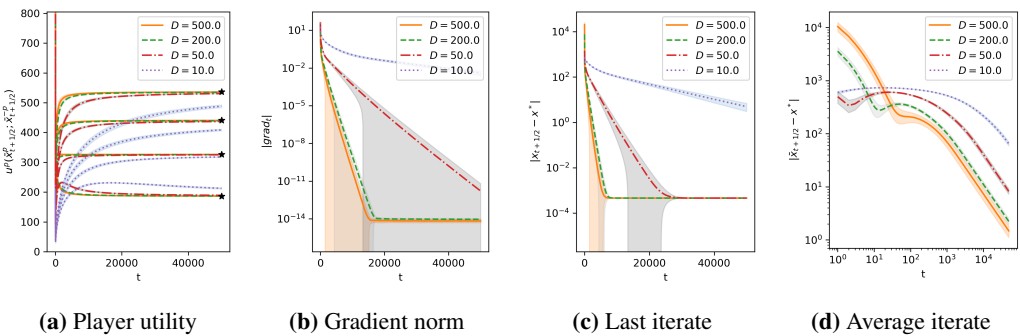

**(a)** Player utility  **(b)** Gradient norm  **(c)** Last iterate  **(d)** Average iterate

**Figure H.5:** We compare the effect of the diameter choice for adaptive DualX on a Kelly auction with $\sigma_{\text{rel}} = 0.2$. We can observe the $\mathcal{O}(1/T)$ average iterate rate for the whole spectrum of diameters but we note that the choice of diameter still has practical impact on convergence time. The fastest convergence is achieved with the highest diameter but note that the method did not converge for $D = 1000$ which we have excluded to keep the plots readable.

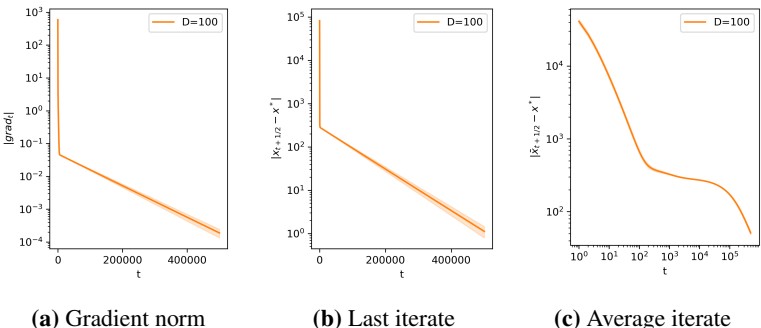

**(a)** Gradient norm      **(b)** Last iterate      **(c)** Average iterate

**Figure H.6:** DualX in a higher dimensional kelly auction ($N = 100$). Let the total resources be $Q = 1000$, cost of biddding be $Z = 100$ and the marginal utility gains be $G = (6.001, 6.002, ..., 6.1)$ (see Section 6 for exact definitions). Since the problem size is out of scope for Mathematica to provide a numerical solution we computed the optimal point using adaptive DualX for 2.5 million full steps with fine-tuned diameter in a deterministic setting. The experiment is then subsequently performed with $\sigma_{\text{rel}} = 0.1$ for 500 000 iterations.

## H.2 Learning a covariance matrix

We apply adaptive DualX to the non-convex problem of learning a covariance matrix introduced in [12] (see Fig. H.9). To fit our unconstrained setting we avoid weight clipping. Thus for fair comparison we include trajectories of GD and OG as well, under these different conditions (see Fig. H.7 and Fig. H.8 respectively). The experiments builds on the code provided by the authors in under the MIT license [12].

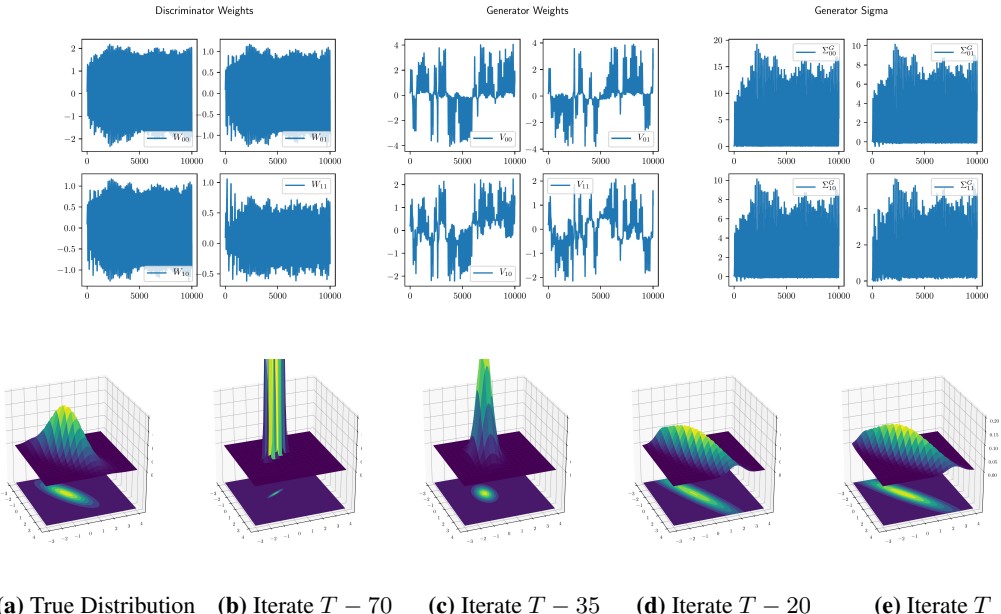

**(a)** True Distribution    **(b)** Iterate $T - 70$    **(c)** Iterate $T - 35$    **(d)** Iterate $T - 20$    **(e)** Iterate $T$

**Figure H.7:** GD for covariance learning of a two-dimensional gaussian without weight clipping using a batch size of 50. Comparison of true distribution and distribution of generator at various points at the end of training.

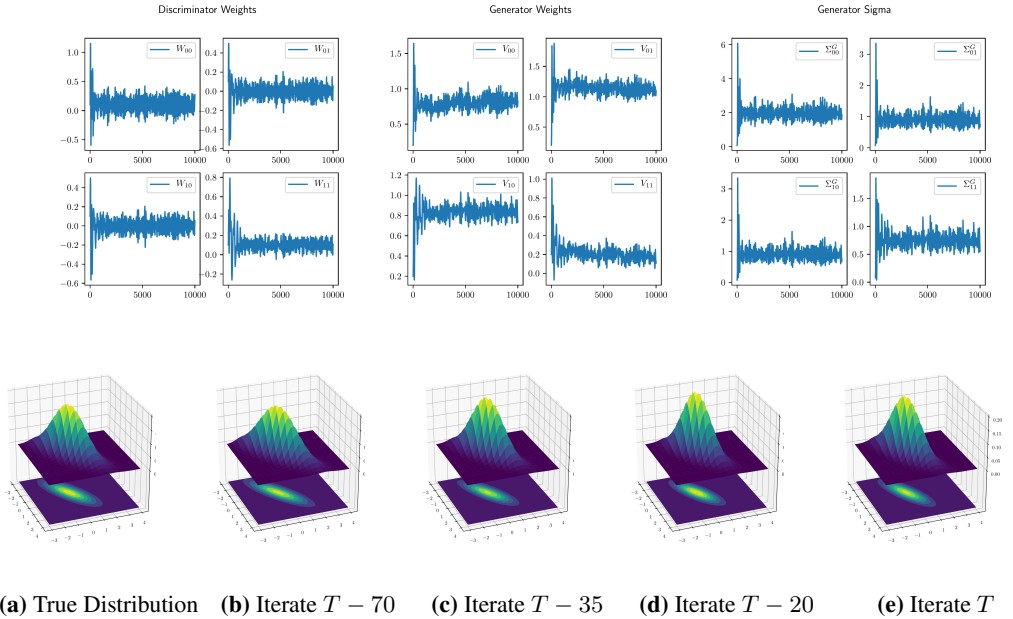

**(a)** True Distribution **(b)** Iterate $T - 70$ **(c)** Iterate $T - 35$ **(d)** Iterate $T - 20$ **(e)** Iterate $T$

**Figure H.8:** OG for covariance learning of a two-dimensional gaussian without weight clipping using a batch size of 50. Comparison of true distribution and distribution of generator at various points at the end of training.

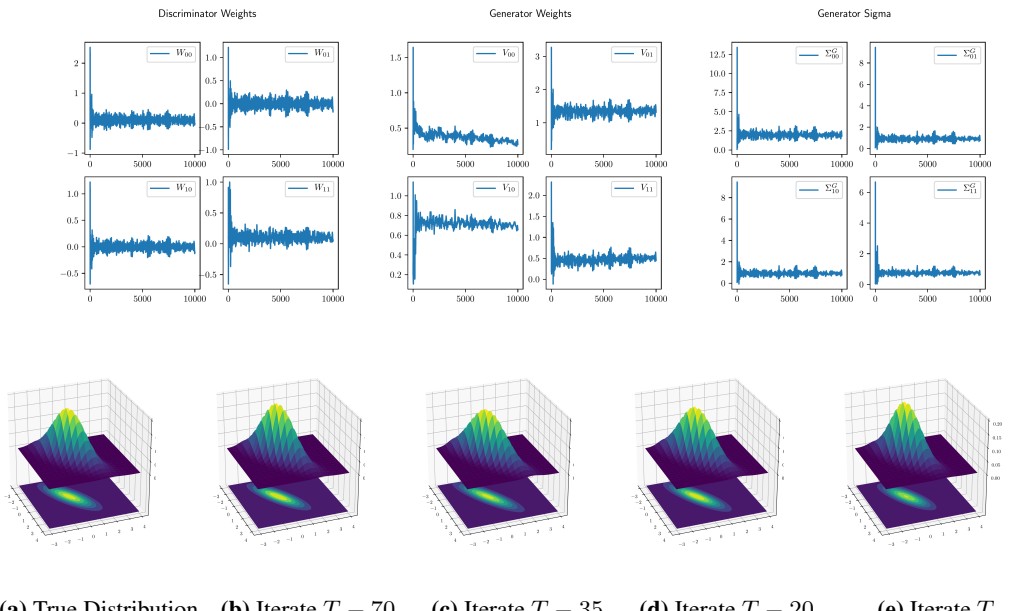

**(a)** True Distribution **(b)** Iterate $T - 70$ **(c)** Iterate $T - 35$ **(d)** Iterate $T - 20$ **(e)** Iterate $T$

**Figure H.9:** Adaptive DualX for covariance learning of a two-dimensional gaussian without weight clipping using a batch size of 50. Comparison of true distribution and distribution of generator at various points at the end of training.