# OpenReview forum: "Sifting through the noise: Universal first-order methods for stochastic variational inequalities"
_NeurIPS.cc/2021/Conference — NeurIPS 2021 Poster_

### Official Review · Reviewer_7yTa · 2021-07-11

**Rating:** 6
**Confidence:** 3

**Summary:**

In this paper, the authors studied solving for variational inequality with monotone cocoercive operator. Two noise models on the oracle are introduced, and a generalized extra-gradient (GEG) family of algorithms are analyzed, encompassing some known algorithms in the literature. In the case of non-adaptively chosen stepsize, convergence rates of $O(1/\sqrt{T})$ and $O(1/T)$ are given for the two noise models. Another class of adaptive-stepsize schemes (somewhat resembling Adagrad type of choice) are studied that can automatically adjust for the stepsize to achieve similar type of convergence without knowledge of the noise structure. Numerical experiments are also conducted on a set of synthetic examples.

**Limitations And Societal Impact:**

Yes.

**Main Review:**

The relevant literature is adequately surveyed, and the paper is clearly written with the nice framework cleanly abstracted. Regarding the adaptive vs. non-adaptive stepsize schedule and the theorem statement, it would be good to explicitly specify the dependence on the lipschitz constant/noise variance etc., instead of focusing only on the dependence on $T$ for a fairer comparison. In a different vein, how would line-search type method compare against these adaptive procedures?

The claims do look right. But it is not clear to me when does the noise model in Assumption 3 hold in practice or if one could check/come up with an implementable procedure to ensure such a condition (but perhaps a control-variate based idea from stochastic optimization could help?) It would be nice if the authors could justify such an assumption. The analysis itself is reminiscent of those stochastic-(sub)gradient based algorithm from optimization.

Minor typos: line 223: sttep-sizes -> step-sizes; line 233: thhe sequel -> the sequel

============== post author-response ===============
I thank the authors for addressing my comments and would like to keep my score unchanged.


**Time Spent Reviewing:**

3

---

> ### Author Response · Authors · 2021-08-10
> **Replies to reviewer**
>
> We thank the reviewer for their thoughtful comments and positive evaluation! We reply below to the points the raised one-by-one:
>
> 1. On the practicality of the relevant noise assumption. There are several generic practical scenarios where this assumption arises naturally. We discuss below one such example from convex optimization and another from game theory:
>     - Random coordinate descent (RCD). Consider a smooth convex function $f$ over $\mathbb{R}^d$, as per Example 1 in the paper. Then, the RCD algorithm draws one coordinate $i_t$ at random at each stage, and calculates the partial derivative $v_{i,t} = \partial f / \partial x_{i_t}$. Subsequently, the $i$-th derivative is updated as $X_{i,t+1} = X_{i,t} - d \gamma_t v_{i,t}$.
>         This update rule can be written in abstract recursive form as
>
>         $$x^{+} = x - \gamma \hat v(x)$$
>
>         where $\hat v_i(x) = d \partial f / \partial x_i  \hat z$ and $\hat z$ is drawn uniformly at random from the set of basis vectors $\{e_1,\dotsc,e_d\}$ of $\mathbb{R}^d$. Clearly, $\mathbb{E}[\hat v(x)] = \nabla f(x)$ by construction; moreover, since $\partial f / \partial x_i = 0$ at the minimum points of $f$, we also have $\hat v(x^{\ast}) = 0$ whenever $x^{\ast}$ is a minimizer of $f$ – i.e., the variance of the estimator $\hat v$ vanishes at the minimum points of $f$. In particular, it is straightforward to verify that $\mathbb{E}[\|\hat v(x) - \nabla f(x)\|^2] = \mathcal{O}(\|\nabla f(x)\|^2)$, which is precisely the relative noise condition for the operator $A = \nabla f$.
>
>      - Random player updating. Consider an $N$-player convex game with loss functions $f_i$, $i=1,\dots,,N$, as per Example 3. Assume further that, at each stage, player $i$ is selected with probability $p_i$ to play an action, and they follow the individual gradient descent rule
>
>         $$X_{i,t+1} = X_{i,t} + \gamma_{t}/p_{i} V_{i,t}$$
>
>         where $V_{i,t} = \nabla_i f_i(X_t)$ denotes the player's individual gradient at the state $X_t = (X_{1,t},...,X_{N,t})$, and $p_i$ is included for scaling reasons.
>
>         It is then easy to see that $\mathbb{E}[V_t] = A(X_t)$ where, $A_i(x) = \nabla_i f_i(x)$. Working as in the case of randomized coordinate descent, it is straightforward to show that $V_t$ is an unbiased oracle for $A$, and since all individual components of $A$ vanish at the game's Nash equilibria, it is also straightforward to verify that $V_t$ satisfies the relative noise condition.
>
>     We will of course be happy to add a detailed discussion of these examples in the main body of our paper.
>
> 2. On the dependence on the problem's parameters. The precise dependence of the bounds is illustrated in the respective proofs provided in the paper's supplement. However, we agree with the reviewer that this is tiresome for the reader to dig through, so we shall insert these explicit bounds in the main part of the paper as well.
>
> 3. On the use of line-search methods. It is not clear if and how a line search could be beneficial in a stochastic setting. Specifically, a line search – Armijo-like or otherwise – would require intermediate operator evaluations that are random, and hence could incur a negative effect on the search process. Moreover, even if the same instantiation is used for all the search steps (e.g., if the random seed is controlled by the optimizer and is forced to be the same throughout the search process), this would constitute an optimized search along a potentially erroneous direction (as indicated by the randomness of the oracle). This is one of the main reasons that line search methods are not very widely used in stochastic environments. We will of course be happy to include a remark about this in the main body of our paper, but we are not otherwise aware of any method handling either of these issues satisfactorily.
>
> We hope and trust that the above addresses the points you raised – and we will of course incorporate all of the above remarks in our revision. Thank you again for your positive evaluation!

---

### Official Review · Reviewer_cKdA · 2021-07-13

**Rating:** 6
**Confidence:** 4

**Summary:**

This paper focuses on the variational inequality problem and proposes a framework to solve this types of problems. The paper further analyses its performance under different noise assumptions and obtain convergence rates for a wide range of algorithms.

**Limitations And Societal Impact:**

The potential negative societal impact is not a must for this work.

I don't have any major problem for this paper. There are only some minor problems:

I am not impressed by the relative randomness part. I don't think this is a very useful assumption (i.e. the randomness vanishes around the optimizer). And I am not surprised to see that this assumption can give $O(\frac{1}{T})$ convergence rate as the oracle case.

As far as I understand, the Assumption 2.1 (almost sure boundedness) is not necessary for non-adaptive step size. Those results in Section 4 hold without it. It only affects Section 5. If this is the case, the authors can add one or two sentences to comment on it.

In many places, the paper use "$<\infty$" to represent two different (although related) meanings. One is that a random variable or a series does not diverge, which is the standard meaning. Another is that a constant which is not depend on $T$, which is very important for convergence analysis. The authors should explain the usage briefly, or change to a different notation, to avoid any confusion.

Line 171, missing reference.

Line 496, $X$ should be $x$.

Line 252, Line 584 - Line 605, some $G$ should be $M$.

Line 587, the middle two steps are unnecessary.

Line 603, missing right bracket.

Line 643, the sentence is meaningless.

**Main Review:**

I don't see any major flaws in the proofs. The presentations are very clear. The technical part is strong.

Although the paper does not propose a new algorithm for variational inquality problem, it successfully unifies previous existing analysis and can be potentially used to find new algorithms.

**Time Spent Reviewing:**

4

---

> ### Author Response · Authors · 2021-08-10
> **Reply to reviewer**
>
> Thanks a lot for your detailed comments and the positive evaluation of our paper! We provide below a point-to-point reply to your questions and remarks:
>
> 1. On the practicality of the relative noise assumption. There are several generic practical scenarios where this assumption arises naturally. We discuss below one such example from convex optimization and another from game theory:
>
>     - Random coordinate descent (RCD). Consider a smooth convex function $f$ over $\mathbb{R}^d$, as per Example 1 in the paper. Then, the RCD algorithm draws one coordinate $i_t$ at random at each stage, and calculates the partial derivative $v_{i,t} = \partial f / \partial x_{i_t}$. Subsequently, the $i$-th derivative is updated as $X_{i,t+1} = X_{i,t} - d \gamma_t v_{i,t}$.
>
>         This update rule can be written in abstract recursive form as
>
>         $$x^{+} = x - \gamma \hat v(x)$$
>
>         where $\hat v_i(x) = d \partial f / \partial x_i  \hat z$ and $\hat z$ is drawn uniformly at random from the set of basis vectors $\{e_1,\dotsc,e_d\}$ of $\mathbb{R}^d$. Clearly, $\mathbb{E}[\hat v(x)] = \nabla f(x)$ by construction; moreover, since $\partial f / \partial x_i = 0$ at the minimum points of $f$, we also have $\hat v(x^{\ast}) = 0$ whenever $x^{\ast}$ is a minimizer of $f$ – i.e., the variance of the estimator $\hat v$ vanishes at the minimum points of $f$. In particular, it is straightforward to verify that $\mathbb{E}[\|\hat v(x) - \nabla f(x)\|^2] = \mathcal{O}(\|\nabla f(x)\|^2)$, which is precisely the relative noise condition for the operator $A = \nabla f$.
>
>     - Random player updating. Consider an $N$-player convex game with loss functions $f_i$, $i=1,\dots,,N$, as per Example 3. Assume further that, at each stage, player $i$ is selected with probability $p_i$ to play an action, and they follow the individual gradient descent rule
>
>         $$X_{i,t+1} = X_{i,t} + \gamma_{t}/p_{i} V_{i,t}$$
>
>         where $V_{i,t} = \nabla_i f_i(X_t)$ denotes the player's individual gradient at the state $X_t = (X_{1,t},...,X_{N,t})$, and $p_i$ is included for scaling reasons.
>
>         It is then easy to see that $\mathbb{E}[V_t] = A(X_t)$ where, $A_i(x) = \nabla_i f_i(x)$. Working as in the case of randomized coordinate descent, it is straightforward to show that $V_t$ is an unbiased oracle for $A$, and since all individual components of $A$ vanish at the game's Nash equilibria, it is also straightforward to verify that $V_t$ satisfies the relative noise condition.
>
>     We will of course be happy to add a detailed discussion of these examples in the main body of our paper.
>
> 2. On lighter assumptions for non-adaptive step-sizes. The reviewer  is correct that lighter assumptions are required for the non-adaptive setting: in particular, instead of a bounded support, we can assume boundedness in expectation. We chose the blanket assumption of bounded support to provide a more uniform presentation, but we will gladly adjust the non-adaptive statements accordingly.
>
> 3. On the notation “$<\infty$”. We will clarify the relevant instances, thanks for pointing this out.
>
> 4. Minor remarks and typos. We will of course fix these, thanks for spotting them!
>
> We hope and trust that the above addresses your remarks – thank you again for your thoughtful review and positive input!

---

> > ### Comment · Reviewer_cKdA · 2021-08-18
> > **Reply to the authors**
> >
> > Thanks for your detailed explanation! I intend to accept the paper and I am happy to adjust my score to 7.

---

### Official Review · Reviewer_Nv5u · 2021-07-15

**Rating:** 7
**Confidence:** 4

**Summary:**

This paper proposes universal adaptive stochastic gradient-based methods for variational inequalities.  There are at least three remarkable contributions. Firstly, it unifies several popular dual averaging algorithms, including stochastic dual averaging, stochastic dual extrapolation, and stochastic optimistic dual averaging. Secondly, with an AdaGrad-type stepsize that is parameter-free, the algorithms achieve an $O(1/\sqrt{T})$ complexity (for the restricted gap) when the stochastic gradients have absolutely bounded variances and an $O(1/T)$ complexity when the the stochastic gradients have relatively bounded variances, without tweaking the hyperparameters. This gives universal and adaptive algorithms. Lastly, the paper shows the asymptotic convergence of the last iterate, which is usually the output in practice.

**Limitations And Societal Impact:**

My comments are mainly about the notation inconsistency and minor issues.

(1) The cocoercivity parameter is denoted as $\beta$ in equation (CC), but $1/L$ in Theorem 2 and proofs.

(2) The bound for $g(x, \omega)$ is denoted as $M$ in Assumption 2 and 3, but $G$ is equation (19) and proofs, if I understand correctly.

(3) I believe the first part of Proposition 1 is weaker than what is proved in Appendix. The proof shows that if $\hat{x}$ is a solution, then $\text{Gap}_{\mathcal{C}}(\hat{x}) = 0$. The current version in Proposition 1 is only an intermediate step.

(4) Above equation (F.11): "Assume that $\beta_{\infty} = +\infty$" should be "Assume that $\sum_{t=1}^{\infty}\beta_t = \infty$ with non-zero probability.

(5) Could you add a short proof on why the vector field in Section 6 is cocoercive? It would be informative to get a sense of how the cocoercivity parameter depends on Q and Z.

(6) I would suggest adding the dependence on $L$ and $G$ in those rates. It seems that the $O(\sqrt{T})$ rate does not even require cocoercivity.

(7) Appendix, equation (E.23), first line: extra vertical line in $X_{t+1/2} - X_t |$.

(8) Are there examples other than overparametrization and the one you presented in Section 6 that involve relative random noise?

**Main Review:**

The paper is well-written overall and the technical proofs are clear. It is remarkable that the proposed  algorithms are both universal and adaptive, rendering them much more user-friendly than algorithms with tons of tuning parameters that rely on potentially unknown quantities. In addition, the possibility to adapt to certain noise structures that the work brings up might have broad impact in the optimization community in designing more adaptive algorithms.

**Time Spent Reviewing:**

5

---

> ### Author Response · Authors · 2021-08-10
> **Replies to reviewer**
>
> Thank you very much for your detailed review and interesting questions. We were delighted that you appreciated the technical content and contributions of our paper, and we thank you for your very encouraging remarks.
>
> We reply to your comments one-by-one below:
>
> 1. On examples satisfying the relative noise assumption. Indeed, there are several practical examples of this kind. We discuss below one such example from convex optimization, and another from game theory (and, of course, we will be happy to include them in our revision):
>
>     - Random coordinate descent (RCD). Consider a smooth convex function $f$ over $\mathbb{R}^d$, as per Example 1 in the paper. Then, the RCD algorithm draws one coordinate $i_t$ at random at each stage, and calculates the partial derivative $v_{i,t} = \partial f / \partial x_{i_t}$. Subsequently, the $i$-th derivative is updated as $X_{i,t+1} = X_{i,t} - d \gamma_t v_{i,t}$.
>
>         This update rule can be written in abstract recursive form as
>
>         $$x^{+} = x - \gamma \hat v(x)$$
>
>         where $\hat v_i(x) = d \partial f / \partial x_i  \hat z$ and $\hat z$ is drawn uniformly at random from the set of basis vectors $\{e_1,\dotsc,e_d\}$ of $\mathbb{R}^d$. Clearly, $\mathbb{E}[\hat v(x)] = \nabla f(x)$ by construction; moreover, since $\partial f / \partial x_i = 0$ at the minimum points of $f$, we also have $\hat v(x^{\ast}) = 0$ whenever $x^{\ast}$ is a minimizer of $f$ – i.e., the variance of the estimator $\hat v$ vanishes at the minimum points of $f$. In particular, it is straightforward to verify that $\mathbb{E}[\|\hat v(x) - \nabla f(x)\|^2] = \mathcal{O}(\|\nabla f(x)\|^2)$, which is precisely the relative noise condition for the operator $A = \nabla f$.
>
>     - Random player updating. Consider an $N$-player convex game with loss functions $f_i$, $i=1,\dots,,N$, as per Example 3. Assume further that, at each stage, player $i$ is selected with probability $p_i$ to play an action, and they follow the individual gradient descent rule
>
>         $$X_{i,t+1} = X_{i,t} + \gamma_{t}/p_{i} V_{i,t}$$
>
>         where $V_{i,t} = \nabla_i f_i(X_t)$ denotes the player's individual gradient at the state $X_t = (X_{1,t},...,X_{N,t})$, and $p_i$ is included for scaling reasons.
>         It is then easy to see that $\mathbb{E}[V_t] = A(X_t)$ where, $A_i(x) = \nabla_i f_i(x)$. Working as in the case of randomized coordinate descent, it is straightforward to show that $V_t$ is an unbiased oracle for $A$, and since all individual components of $A$ vanish at the game's Nash equilibria, it is also straightforward to verify that $V_t$ satisfies the relative noise condition.
>
>     We will of course be happy to add a detailed discussion of these examples in the main body of our paper.
>
> 2. The notation $1/L$. This was done at some point because, by the Baillon-Haddad theorem, a function is $L$-Lipschitz smooth if and only if $\nabla f$ is $(1/L)$-cocoercive. This basic result has led to the two notations being used sometimes interchangeably in the literature – we will stick to a single notational convention to avoid any ambiguities.
>
> 3. On the Kelly auction example. Of course, we will add a complete proof (it relies a Jacobian calculation).
>
> 4. On Proposition 1. The reviewer is correct, this was an omission on our part.  We will update statement in the main part to reflect the stronger statement proven in the appendix.
>
> 5. On the dependence on $L$ and $G$. The precise dependence of the derived bounds is illustrated in the respective proofs provided in the paper's supplement. However, we agree with the reviewer that this is tiresome for the reader to dig through, so we shall insert these explicit bounds in the main part of the paper as well.
>
> 6. Typos and minor points. We will of course fix all those, thanks for spotting them!
>
> We hope and trust that the above addresses your remarks – thank you again for your thoughtful review and positive input!

---

> > ### Comment · Reviewer_Nv5u · 2021-08-20
> > **Reply to the authors**
> >
> > I would like to thank the authors for their detailed response. The examples satisfying the relative noise condition are very interesting! I will keep my positive score.

---

### Official Review · Reviewer_negE · 2021-07-17

**Rating:** 7
**Confidence:** 3

**Summary:**

The paper gives an algorithmic framework of generalized extragradient method for solving variational inequalities induced by a cocoercive operator A. The framework casts dual averaging, dual extrapolation as special cases, and interpolates between the convergence rate of O(1/sqrt{T}) and O(1/T) for different noise levels assuming a stochastic oracle of A(x). They also provide an adaptive variant of their methods that obtain the same rate without needing to know the explicit problem parameters. Finally, they corroborated their results by some preliminary experiments to show that the convergence and corresponding rates under different noise levels.

**Limitations And Societal Impact:**

The authors list the limitation of the paper in a clear way in Section 7 when they make the conclusion.

**Main Review:**

Overall the paper is well-structured, well-written and includes a few enlightening results. I have enjoyed reading the paper. In particular, I find the techniques and results the authors showing interesting and novel, in the following senses:

1) The authors propose a nice and clean framework that cast  many known methods as special cases, and fully characterize the convergence rate of their method under two commonly-seen noisy oracle models for solving VIs, under the cocoercitivity assumption. I have not seen many 1/T results, or last-iterate converging result under noisy oracles for such problems, and think this is a very interesting result under its own.

2) The authors also give methods with adaptive stepsize choices that doesn't depend on problem parameters, and also provide theoretical support of the same convergence rate. This makes a further step toward improving the practicality of the methods, which many newly-designed first-order methods for such problems fall short of.

3) The flow of the paper is easy to follow. Especially while the authors explain how they achieve the results, they show their key lemma is a summable property of the oracle return terms, which in turn relies crucially on the cocoercive property of the operator. I think this will help readers understand their techniques better and might facilitate interesting progress in other relevant problems.

Considering all these factors, I think it is a nice paper worth sharing in the community. The only suggestion of mine would be to motivate the consideration of relatively noisy oracles a bit more, perhaps in the specific context of examples 1-3 raised in the paper.

* minor comment: line 122 "relative random..." looks like an unfinished sentence.

** Update: I have read the authors' responses and my concerns are addressed properly. Based on these, I am happy to keep my score.

**Time Spent Reviewing:**

4

---

> ### Author Response · Authors · 2021-08-10
> **Reply to reviewer**
>
> Thank you for your thoughtful comments, encouraging remarks and positive evaluation!
>
> Regarding your comment about the relative noise assumption, consider the following further examples where oracles of this type arise naturally (one from convex optimization, another from game theory):
>
> 1. Random coordinate descent (RCD). Consider a smooth convex function $f$ over $\mathbb{R}^d$, as per Example 1 in the paper. Then, the RCD algorithm draws one coordinate $i_t$ at random at each stage, and calculates the partial derivative $v_{i,t} = \partial f / \partial x_{i_t}$. Subsequently, the $i$-th derivative is updated as $X_{i,t+1} = X_{i,t} - d \gamma_t v_{i,t}$.
>
>     This update rule can be written in abstract recursive form as
>     $$x^{+} = x - \gamma \hat v(x)$$
>
>     where $\hat v_i(x) = d \partial f / \partial x_i  \hat z$ and $\hat z$ is drawn uniformly at random from the set of basis vectors $\{e_1,\dotsc,e_d\}$ of $\mathbb{R}^d$. Clearly, $\mathbb{E}[\hat v(x)] = \nabla f(x)$ by construction; moreover, since $\partial f / \partial x_i = 0$ at the minimum points of $f$, we also have $\hat v(x^{\ast}) = 0$ whenever $x^{\ast}$ is a minimizer of $f$ – i.e., the variance of the estimator $\hat v$ vanishes at the minimum points of $f$. In particular, it is straightforward to verify that $\mathbb{E}[\|\hat v(x) - \nabla f(x)\|^2] = \mathcal{O}(\|\nabla f(x)\|^2)$, which is precisely the relative noise condition for the operator $A = \nabla f$.
>
> 2. Random player updating. Consider an $N$-player convex game with loss functions $f_i$, $i=1,...,N$, as per Example 3. Assume further that, at each stage, player $i$ is selected with probability $p_i$ to play an action, and they follow the individual gradient descent rule
>
>     $$X_{i,t+1} = X_{i,t} + \gamma_{t}/p_{i} V_{i,t}$$
>
>      where $V_{i,t} = \nabla_i f_i(X_t)$ denotes the player's individual gradient at the state $X_t = (X_{1,t},...,X_{N,t})$, and $p_i$ is included for scaling reasons.
>
>     It is then easy to see that $\mathbb{E}[V_t] = A(X_t)$ where, $A_i(x) = \nabla_i f_i(x)$. Working as in the case of randomized coordinate descent, it is straightforward to show that $V_t$ is an unbiased oracle for $A$, and since all individual components of $A$ vanish at the game's Nash equilibria, it is also straightforward to verify that $V_t$ satisfies the relative noise condition.
>
> We will be happy to include these examples in detail in our revision – and, of course, we will also correct the small typo you pointed out.
>
> We hope that the above answers your questions, and we will be sure to include them in our revision. Thank you again for your thoughtful review and positive input!

---

### Decision · Program_Chairs · 2021-09-27

**Decision:**

Accept (Poster)

**Comment:**

This paper proposes universal adaptive stochastic gradient-based methods for variational inequalities. It unifies several dual averaging algorithms, uses AdaGrad-type stepsize to achieve universality and adaptive results, and shows asymptotic convergence of the last iterate. I agree with the reviewers that the technical contributions are interesting and novel, and I am happy to recommend acceptance.